# Change Point Detection in the Frequency Domain with Statistical Reliability

**Akifumi Yamada**                                    *yamada.akifumi.nagoyaml@gmail.com*
*Nagoya University*

**Tomohiro Shiraishi**                              *shiraishi.tomohiro.nagoyaml@gmail.com*
*Nagoya University*
*RIKEN*

**Shuichi Nishino**                                  *nishino.shuichi.nagoyaml@gmail.com*
*Nagoya University*
*RIKEN*

**Teruyuki Katsuoka**                              *katsuoka.teruyuki.nagoyaml@gmail.com*
*Nagoya University*

**Kouichi Taji**                                              *taji@nagoya-u.jp*
*Nagoya University*

**Ichiro Takeuchi**                              *takeuchi.ichiro.n6@f.mail.nagoya-u.ac.jp*
*Nagoya University*
*RIKEN*

**Reviewed on OpenReview:** *https://openreview.net/forum?id=FNRdaHz3qN*

## Abstract

Effective condition monitoring in complex systems requires identifying change points (CPs) in the frequency domain, as the structural changes often arise across multiple frequencies. This paper extends recent advancements in statistically significant CP detection, based on Selective Inference (SI), to the frequency domain. The proposed SI method quantifies the statistical significance of detected CPs in the frequency domain using $p$-values, ensuring that the detected changes reflect genuine structural shifts in the target system. We address two major technical challenges to achieve this. First, we extend the existing SI framework to the frequency domain by appropriately utilizing the properties of discrete Fourier transform (DFT). Second, we develop an SI method that provides valid $p$-values for CPs where changes occur across multiple frequencies. Experimental results demonstrate that the proposed method reliably identifies genuine CPs with strong statistical guarantees, enabling more accurate root-cause analysis in the frequency domain of complex systems.

## 1 Introduction

To detect failures in complex systems, it is crucial to accurately identify the frequencies where the significant changes occur. By pinpointing these specific frequencies, the root causes of faults in the systems can be identified. In this paper, we address the change point (CP) detection problem in the frequency domain and propose a method that can identify *statistically significant* changes in specific frequencies. Quantifying statistical significance in CP detection ensures that the detected changes in the frequency domain reflect genuine structural alterations rather than random noise. Measures such as $p$-values provide a quantitative framework to differentiate real changes from spurious ones, effectively mitigating the risk of incorrect decisions in data-driven systems.

This study was motivated by recent developments in statistically significant CP detection based on *Selective Inference (SI)* (Taylor & Tibshirani, 2015; Fithian et al., 2015; Lee & Taylor, 2014). Prior to SI, quantifying the significance of detected CPs was challenging due to the issue of *double dipping*, i.e., using the same data to both identify and test CPs inflates false positive findings. Traditional statistical approaches prior to SI primarily focused on testing only whether a CP exists or not within a certain range using multiple testing framework based on asymptotic distribution (Page, 1954; Mika et al., 1999). Therefore, providing statistical significance measures, such as $p$-values, for specific points or frequencies was not feasible. SI is a novel statistical inference framework designed for data-driven hypotheses. In the context of CP detection, it enables the evaluation for the statistical significance of detected CPs conditional on a specific CP detection algorithm, thereby resolving the aforementioned double dipping issue. The use of SI for statistically significant CP detection in the time domain has been actively studied recently (Hyun et al., 2018; Duy et al., 2020). Our contribution in this study is to extend these methods and realize statistically significant CP detection in the frequency domain.

One of the difficulties in frequency-domain CP detection is that changes often appear across multiple frequencies due to shared underlying phenomena influencing broad signal characteristics. Figure 1 shows an example of frequency-domain CP detection problems along with the statistically significant CPs identified using our proposed SI method. Panel (a) shows a time series transformed into signals across five frequencies. Panel (b) presents the CP detection results for each individual frequency $d \in \{d_0, d_1, d_2, d_3, d_4\}$ [1]. Panel (c) integrates CPs across multiple frequencies, detecting three changes A, B, and C (at CP A, changes occur in frequencies $d_1$ and $d_2$; at CP B, in $d_2$ and $d_3$; and at CP C, only in $d_0$). In table (d), the obtained $p$-values for changes A, B, and C using the proposed method are 0.008, 0.006, and 0.752 (denoted as *selective p-values*), respectively, indicating that at a significance level of $\frac{0.05}{3} \approx 0.0167$ decided by Bonferroni correction, changes A and B are statistically significant CPs[2].

To perform statistically significant CP detection, as illustrated in Figure 1, two key technical challenges must be addressed. The first challenge is to extend the SI framework to the frequency domain. We tackle this by formulating the test statistic and the conditioning on the hypothesis selection that accurately account for the properties of discrete Fourier transform (DFT). The second challenge involves quantifying the statistical significance of CPs shared across multiple frequencies. Addressing this requires solving a combinatorial optimization problem, which is computationally infeasible to solve globally optimally, necessitating a heuristic approach to obtain the approximate solution. In this study, we employ a method based on simulated annealing (SA) (Kirkpatrick et al., 1983; Černý, 1985) and implement the SI framework to appropriately quantify the statistical significance of the approximate solution derived from the heuristic algorithm.

The proposed CP detection method consists of two stages. In the first stage, CP candidates in the frequency domain are selected. Since the selection of these CP candidates is formulated as a combinatorial optimization problem, a heuristic algorithm is used to derive an approximate solution. In the second stage, the statistical significance of each CP candidate selected in the first stage is quantified in the form of $p$-values using the SI framework. Among the CP candidates, only those with $p$-values below a significance level (e.g., 0.05 or 0.01) are eventually detected as final CPs. The probability of the final detected CPs being false positives is theoretically guaranteed to be below the specified significance level.

The rest of this paper is organized as follows. Section 2 formulates the problem. Section 3 explains the heuristic algorithm used to select CP candidates in the first stage of the proposed method. Section 4 describes the method for quantifying the statistical significance of CP candidates using the SI framework in the second stage. In Section 5, we demonstrate the effectiveness of our proposed method through comprehensive numerical experiments using both synthetic and real-world data. For reproducibility, our implementation is available at `https://github.com/Takeuchi-Lab-SI-Group/si_for_frequency-domain_change_point_detection`. Finally, Section 6 concludes this paper.

---

[1] Note that while the "amplitude" spectra are presented in the figures of this paper to visualize the temporal variations of spectral sequences and the means for each segment, the "complex" spectra are utilized in the actual CP detection and the hypothesis testing (see Section 2 and the subsequent sections for details).

[2] In Figure 1, the values labeled as naive $p$-values are inappropriate because they fail to account for double-dipping, leading to excessively small values and an inflated rate of false positive findings.

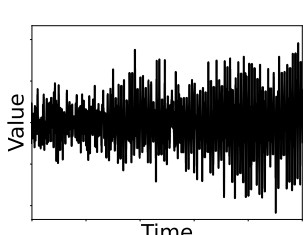

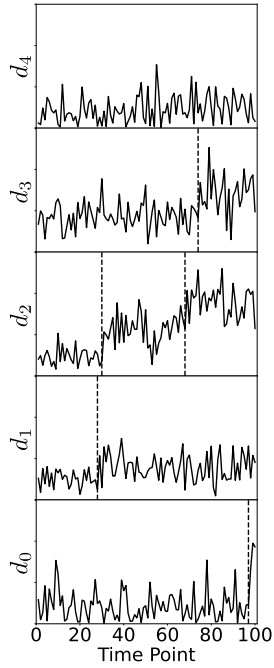

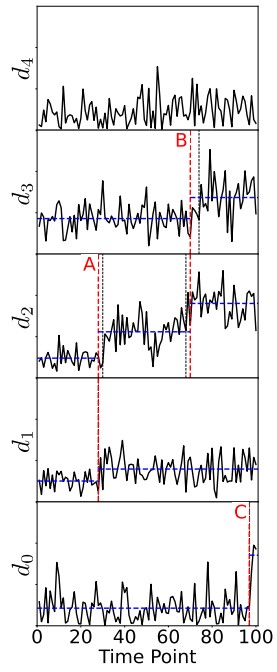

(a) Time series signal.

(b) CP detection result for each frequency component.

(c) CP detection result after integrating CPs across multiple frequencies.

(d) Statistical test for the CPs detected in (c).

| Time Point | A | B | C |
|---|---|---|---|
| Detection | True | True | Wrong |
| Proposed $p$-value | 0.008 | 0.006 | 0.752 |
| Naive $p$-value | 0.000 | 0.000 | 0.000 |

Figure 1: Demonstration of the proposed method. Panel (a) shows the original time series signal. Panel (b) illustrates the result of CP detection for each time variation of the five frequency components, and panel (c) represents the CPs merged across multiple frequencies. In panel (c), the CPs for frequency $d_1$ at time point 28 and $d_2$ at 30, and those for $d_2$ at 68 and $d_3$ at 74 merge into one CP at 28 (A) and 70 (B), respectively. Actually, the CPs A and B are truely detected, while a CP C for $d_0$ at 97 is wrong detection. Table (d) shows the results of statistical test for the CPs detected in (c). The proposed $p$-value is enough large for falsely detected CP C, while the naive metod causes a false positive because the $p$-value is too small. Furthermore, the proposed method provides sufficiently small $p$-values for truely detected CPs A and B, which are statistically significant.

**Our contribution.** The main contribution of this study is the development of a statistical testing method, based on the framework of SI, to quantify the statistical significance of detected CPs in the frequency domain. While CP detection in the frequency domain is an important problem in engineering and several algorithms have been proposed, to the best of our knowledge, no existing method properly quantifies the statistical significance of CPs in the form of $p$-values. By leveraging the recently advancing SI framework, we provide a solution to this unresolved issue. There are various CP detection algorithm in the frequency domain. In this study, as an example, we consider a CP detection algorithm that identifies changes common across multiple frequencies using dynamic programming (DP) and simulated annealing. It should be noted that our contribution does not lie in this CP detection algorithm itself, but in proposing a statistical test to quantify the significance of the detected CPs.

**Related work.** The CP detection problem has long been studied with various applications in a variety of fields, such as finance (Fryzlewicz & Subba Rao, 2014; Pepelyshev & Polunchenko, 2017), bioinformatics (Chen & Wang, 2008; Muggeo & Adelfio, 2011; Pierre-Jean et al., 2015), climatology (Reeves et al., 2007; Beaulieu et al., 2012), and machine monitoring (Lu et al., 2017; 2018). In the statistics and machine learning communities, various methods have been proposed for identifying multiple CPs from univariate sequences. The most straightforward approach is repeatedly applying single CP detection algorithms, such as binary segmentation (Scott & Knott, 1974). Examples of such approaches include circular binary segmentation (Olshen et al., 2004) and wild binary segmentation (Fryzlewicz, 2014). Another line of research has proposed numerous approaches based on penalized likelihood, such as Segment Neighbourhood (Auger & Lawrence, 1989), Optimal Partitioning (Jackson et al., 2005), PELT (Killick et al., 2012), and FPOP (Maidstone et al., 2017). In these studies, dynamic programming is employed to solve the penalized likelihood minimization problem. While most CP detection studies focus on the time domain, a few studies have targeted the frequency domain, such as Adak (1998), Last & Shumway (2008), and Preuss et al. (2015).

Most existing studies for statistical inference on detected CPs have relied on asymptotic theory under restrictive assumptions, such as weak dependency. For single CP detection problems, approaches such as the CUSUM score (Page, 1954), Fisher discriminant score (Mika et al., 1999; Harchaoui et al., 2009), and MMD (Li et al., 2015) conduct statistical inference on the detected CPs based on asymptotic distribution of some discrepancy measures. For multiple CP detection problems, methods such as SMUCE (Frick et al., 2014) and the MOSUM procedure (Eichinger & Kirch, 2018) also employ asymptotic inference. However, these methods primarily focus on testing whether a CP exists within a certain range rather than directly quantifying the statistical significance of the location of the CP itself. Furthermore, asymptotic approaches often fail to control the false positive (type I error) rate effectively, or resulting in conservative testing with low statistical power (Hyun et al., 2018).

SI was initially introduced as a method of statistical inference for feature selection in linear models (Taylor & Tibshirani, 2015; Fithian et al., 2015) and later extended to various feature selection algorithms, including marginal screening (Lee & Taylor, 2014), stepwise feature selection (Tibshirani et al., 2016), and Lasso (Lee et al., 2016). The core concept of SI is to derive the exact null distribution of the test statistic conditional on the hypothesis selection event, thereby enabling valid statistical inference with controlled type I error rate. Recently, significant attention has been given to applying SI to more complex supervised learning algorithms, such as kernel models (Yamada et al., 2018), boosting (Rügamer & Greven, 2020), tree-structured models (Neufeld et al., 2022), and neural networks (Duy et al., 2022; Miwa et al., 2023; Shiraishi et al., 2024b). Furthermore, SI has proven valuable for unsupervised learning tasks, including clustering (Lee et al., 2015; Chen & Witten, 2023; Gao et al., 2024), outlier detection (Chen & Bien, 2020; Tsukurimichi et al., 2022), domain adaptation (Duy et al., 2024), and segmentation (Tanizaki et al., 2020; Duy et al., 2022). SI was first used for CP detection problem in Hyun et al. (2018), which focused on Fused Lasso algorithm. Since then, the framework has been explored in various CP detection algorithms, including the CUSUM-based method (Umezu & Takeuchi, 2017), binary segmentation and its variants (Hyun et al., 2021), dynamic programming (Duy et al., 2020), and other related problems (Sugiyama et al., 2021; Jewell et al., 2022; Carrington & Fearnhead, 2024; Shiraishi et al., 2024a). Existing studies on CP detection have focused on the time domain, making this the first to offer valid statistical inferences for frequency-domain CPs using the SI framework.

## 2 Problem Setup

In this section, we first desrcribe the probabilistic model of time series on which the statistical inference is based, and then formulate the problem of CP detection in the frequency domain.

### 2.1 Probabilistic Model for Time Series Data

For statistical inference, we interpret that the observed time series data is a realization of a random sequence following a certain probabilistic model. Let us denote the univariate random sequence with length $N$ by

$$\boldsymbol{X} = (X_1, \ldots, X_N)^\top = \boldsymbol{s} + \boldsymbol{\epsilon}, \, \boldsymbol{\epsilon} \sim \mathcal{N}(\boldsymbol{0}, \sigma^2 I_N), \tag{1}$$

where $\boldsymbol{s} \in \mathbb{R}^N$ is the unknown true signal vector, and $\boldsymbol{\epsilon} \in \mathbb{R}^N$ is the normally distributed noise vector with the covariance matrix $\sigma^2 I_N$ [3]. In other words, we assume that the observed time series data is sampled from the probabilistic model $\boldsymbol{X} \sim \mathcal{N}\left(\boldsymbol{s}, \sigma^2 I_N\right)$.

Then, consider applying discrete short-time Fourier transform (STFT) to the random sequence $\boldsymbol{X}$ using rectangular window of width $M$ without overlapping. In this case, the computations of DFT are required $T = \lfloor \frac{N}{M} \rfloor$ times, where we assume $T = \frac{N}{M}$, i.e., $N$ is a multiple of $M$ for simplicity. We denote $M$-point DFT matrix as

$$W_M = \left(\boldsymbol{w}_M^{(0)}, \ldots, \boldsymbol{w}_M^{(M-1)}\right) \in \mathbb{C}^{M \times M},$$

where $\boldsymbol{w}_M^{(d)} = \left(\omega_M^0, \omega_M^d, \ldots, \omega_M^{(M-1)d}\right)^\top$, and $\omega_M = e^{-j\left(\frac{2\pi}{M}\right)}$ for frequency $d \in \{0, \ldots, M-1\}$ in which $j$ is the imaginary unit. Using $\boldsymbol{w}_M^{(d)}$, we consider multiple spectral sequences across $D = \lfloor \frac{M}{2} \rfloor + 1$ frequencies due to the symmetry property of the spectrum. For frequency $d$, the sequence of spectra is written as

$$\boldsymbol{F}^{(d)} = \left(F_1^{(d)}, \ldots, F_T^{(d)}\right)^\top \in \mathbb{C}^T, \, d \in \{0, \ldots, D-1\},$$

where $F_t^{(d)} = \left(\boldsymbol{1}_{t:t} \otimes \boldsymbol{w}_M^{(d)}\right)^\top \boldsymbol{X}$, and $\boldsymbol{1}_{s:e} \in \mathbb{R}^T$ is a vector whose elements from position $s$ to $e$ are set to 1, and 0 otherwise for $1 \le s \le e \le T$.

## 2.2 Statistically Significant CP Detection in the Frequency Domain

As mentioned above, for statistical inference, we interpret that the observed time series is randomly sampled from the probabilistic model in (1), which is denoted by

$$\boldsymbol{x} = (x_1, \ldots, x_N)^\top \in \mathbb{R}^N. \tag{2}$$

Similarly, the sequence of spectra for frequency $d$, obtained by applying the aforementioned STFT to the observed time series $\boldsymbol{x}$, is written as

$$\boldsymbol{f}^{(d)} = \left(f_1^{(d)}, \ldots, f_T^{(d)}\right)^\top \in \mathbb{C}^T, \, d \in \{0, \ldots, D-1\}. \tag{3}$$

The goal of this study is to detect changes in the true signals of frequency spectral sequences $\{\boldsymbol{F}^{(d)}\}_{d \in \{0, \ldots, D-1\}}$ based on the observed sequences $\{\boldsymbol{f}^{(d)}\}_{d \in \{0, \ldots, D-1\}}$.

Among variety of changes, we focus in this paper on *mean-shift* of frequency spectral sequences. Let us denote the mean spectrum of frequency $d$ at time point $t$ by

$$\mu_t^{(d)} = \mathbb{E}[F_t^{(d)}], \, (d, t) \in \{0, \ldots, D-1\} \times [T],$$

where the expectation operator $\mathbb{E}[\cdot]$ is taken with respect to the probabilistic model in (1)[4], and $[T] = \{1, \ldots, T\}$ indicates the set of natural numbers up to $T$.

Considering a segment from time points $s$ to $e$ with $1 \le s \le e \le T$, we say that there is a *mean-shift change* in the frequency $d$ at time point $t \in \{s, \ldots, e-1\}$ if and only if

$$\frac{1}{t-s+1} \sum_{t'=s}^{t} \mu_{t'}^{(d)} \ne \frac{1}{e-t} \sum_{t'=t+1}^{e} \mu_{t'}^{(d)}.$$

As discussed in Section 1, we have prior knowledge that simultaneous changes occur across multiple distinct frequencies. In Section 3, we introduce a heuristic algorithm that generates CP candidates by incorporating this prior knowledge. Subsequently, in Section 4, we present an SI framework to quantify the statistical significance of each CP candidate in the form of $p$-values. Finally, we select the candidates with $p$-values smaller than the user-specified significance level (e.g., 0.05 or 0.01) as our final CPs.

---

[3]In the following discussion, the noise $\boldsymbol{\epsilon}$ must be an independent and identically distributed Gaussian vector with predetermined covariance matrix. The robustness of our proposed method for unknown noise variance, non-Gaussian noise, and correlated noise is discussed in Appendix D.3.

[4]Since the frequency spectral sequences are obtained through linear transformations of Gaussian random variables, the existence of their expectations is guaranteed.

# 3 Heuristic Algorithm for CP Candidate Selection in Multiple Frequencies

As discussed in Section 1, the types of problems occurring in time series signals can be systematically determined by simultaneously detecting CPs across multiple frequencies, such as the harmonics and sidebands of the characteristic frequencies. Thus, we formulate CP candidate selection as an optimization problem that aims to not only estimate the optimal number and location of CPs for each frequency, but also reduce the total number of change locations across all frequencies by aligning the positions.

## 3.1 Objective Function for CP Candidate Selection

Let $K^{(d)}$ be the number of selected CP candidates and $\boldsymbol{\tau}^{(d)} = \{\tau_1^{(d)}, \dots, \tau_{K^{(d)}}^{(d)}\} \subseteq [T-1]$ be the ordered set of CP candidates ($\tau_1^{(d)} < \cdots < \tau_{K^{(d)}}^{(d)}$ and $\tau_0^{(d)} = 0, \tau_{K^{(d)}+1}^{(d)} = T$) for frequency $d \in \{0, \dots, D-1\}$. Furthermore, we define the set of CP candidate locations as

$$\boldsymbol{\tau} = \bigcup_{d=0}^{D-1} \boldsymbol{\tau}^{(d)} = \{\tau_1, \dots, \tau_K\} \subseteq [T-1], \tag{4}$$

where we denote the total number of CP candidate locations as $K = |\boldsymbol{\tau}|$.

Let $\boldsymbol{\mathcal{T}} = (\boldsymbol{\tau}^{(0)}, \dots, \boldsymbol{\tau}^{(D-1)})$ be the collection of all CP candidates across all the $D$ frequencies. Then, given an observed time-series data $\boldsymbol{x}$ in (2), the objective function of our CP candidates $\boldsymbol{\mathcal{T}}$ is written as

$$E(\boldsymbol{\mathcal{T}}, \boldsymbol{x}) = \sum_{d=0}^{D-1} \sum_{k=1}^{K^{(d)}+1} \mathcal{C}\left(\boldsymbol{f}_{\tau_{k-1}^{(d)}+1:\tau_k^{(d)}}^{(d)}\right) + \beta^{(d)} K^{(d)} + \gamma K. \tag{5}$$

The first term of the obejective function in (5) indicates the cost for quantifying the variability of segments between two adjacent CP candidates, where $\mathcal{C}\left(\boldsymbol{f}_{s:e}^{(d)}\right)$ indicates the cost function for a segment $\boldsymbol{f}_{s:e}^{(d)} = (f_s^{(d)}, f_{s+1}^{(d)}, \dots, f_e^{(d)})^\top$ specifically defined as

$$\mathcal{C}\left(\boldsymbol{f}_{s:e}^{(d)}\right) = c_{\text{sym}}^{(d)} \sum_{t=s}^{e} \left| f_t^{(d)} - \frac{1}{e-s+1} \sum_{t'=s}^{e} f_{t'}^{(d)} \right|^2,$$

where

$$c_{\text{sym}}^{(d)} = \begin{cases} 1 & \text{if } d = 0, \frac{M}{2}, \\ 2 & \text{if } d \neq 0, \frac{M}{2}, \end{cases} \tag{6}$$

because the frequency spectra have complex conjugate symmetry. The second term indicates the penalty term for the number of CP candidates in each frequency, while the third term indicate the penalty for the number of total CP candidate locations, where $(\beta^{(0)}, \dots, \beta^{(D-1)}) \in \mathbb{R}^D$ and $\gamma \in \mathbb{R}$ are hyper-parameters for controlling the balance between the three terms (the details of how to determine these hyper-parameters are presented in Appendix A). The third penalty term indicates that we take into account the trade-off between cost and penalty term for not only $K^{(d)}$ but also $K$ in (5), hence CP candidates of multiple frequencies tend to be detected at the same time point.

## 3.2 Approximately Optimizing the Objective Function by Simulated Annealing

Unfortunately, since minimizing the objective function in (5) is a challenging combinatorial optimization problem, we must rely on approximate solutions derived from heuristic algorithms. Following the approach in Lavielle (1998), we employ *simulated annealing* (Kirkpatrick et al., 1983; Černỳ, 1985) to approximately solve the optimization problem. Simulated annealing is a meta-heuristic algorithm widely applied to various practical problems, as it converges asymptotically to a global solution with high probability under specific conditions. To approximately optimize the objective function in (5), we perform the following two steps:

- Step 1: Individually estimate the CP candidates for each frequency. Specifically, we minimize the objective function in (5) with $\gamma = 0$. This can be optimally achieved by applying dynamic programming to each sequence.

- Step 2: Refine the CP candidates for each frequency estimated in step 1 using simulated annealing to estimate the changes shared across multiple frequencies.

The reason for obtaining approximate solutions using simulated annealing is that the global minimization of the objective function in (5) across multiple frequencies cannot be achieved by directly extending the dynamic programming approach used in step 1. This inability to obtain the global optimum stems from the fact that it is impossible to construct a Bellman equation for this optimization problem; specifically, additional information which is not preserved in the solutions of subproblems may be needed at later-stage decisions.

We note that approximately solving the combinatorial optimization problem in (5) is NOT our novel contribution (our key contribution, detailed in Section 4, lies in providing a theoretical guarantee for the false positive detection probability of the obtained approximate solution). For example, a similar approach has been used for analogous problems in studies such as Lavielle (1998). Moreover, we do NOT claim that simulated annealing is the optimal approach for this problem; it is simply one of the reasonable choices, and other meta-heuristics could also be used as alternatives.

### 3.3 Step 1: Generating an Initial Solution by Dynamic Programming

We need to generate an initial solution $\boldsymbol{\mathcal{T}}^{\text{init}} = (\boldsymbol{\tau}^{\text{init}\,(0)}, \ldots, \boldsymbol{\tau}^{\text{init}\,(D-1)})$ before applying simulated annealing. To obtain the initial solution, we first set $\gamma = 0$ in (5) to detect CP candidates for each frequency and formulate an optimization problem as

$$\boldsymbol{\tau}^{\text{init}\,(d)} = \arg\min_{\boldsymbol{\tau}^{(d)}} \sum_{k=1}^{K^{(d)}+1} \mathcal{C}\left(\boldsymbol{f}^{(d)}_{\tau^{(d)}_{k-1}+1:\tau^{(d)}_k}\right) + \beta^{(d)} K^{(d)}. \tag{7}$$

This optimization problem can be solved efficiently using dynamic programming algorithm which is called Optimal Partitioning (Jackson et al., 2005). This method employs the Bellman equation that recursively determines optimal solutions for simpler subproblems. Given $\mathcal{D}^{\text{init}}$ as the set of frequencies for which at least one CP candidate is detected, it is sufficient to apply simulated annealing only to $d \in \mathcal{D}^{\text{init}}$ because introducing an additional CP to any frequency $d$ requires a minimum penalty of $\beta^{(d)}$ as shown in (5). Thus, we reduce the number of optimal solution candidates, and the computational cost of simulated annealing can be decreased without degrading the solution quality.

### 3.4 Step 2: Refining the Solution by Simulated Annealing

We perform multivariate CP candidate selection in the frequency domain using simulated annealing that is applied for solving large combinatorial optimization problems for which finding global optima is difficult. In each step of simulated annealing, the Metropolis algorithm is used to accept transitions not only to improving solutions that decrease the objective function, i.e., $\Delta E(\boldsymbol{\mathcal{T}}', \boldsymbol{\mathcal{T}}, \boldsymbol{x}) = E(\boldsymbol{\mathcal{T}}', \boldsymbol{x}) - E(\boldsymbol{\mathcal{T}}, \boldsymbol{x}) \leq 0$, where $\boldsymbol{\mathcal{T}}$ and $\boldsymbol{\mathcal{T}}'$ are current and new solutions, respectively, but also to deteriorating solutions that increase the objective function, i.e., $\Delta E(\boldsymbol{\mathcal{T}}', \boldsymbol{\mathcal{T}}, \boldsymbol{x}) > 0$, with the probability controlled by a temperature parameter $c$, which allows escape from the local solution. The pseudo code of the Metropolis algorithm is shown in Algorithm 1.

---

**Algorithm 1** `metropolis_algorithm`

---

**Input:** Difference of objective function values $\Delta E$, temperature $c$
1: Uniformly sample $\xi$ from $[0, 1)$
2: **if** $\Delta E + c \ln \xi < 0$ **then**
3:     `status` $\leftarrow$ Acceptance
4: **else**
5:     `status` $\leftarrow$ Rejection
6: **end if**
**Output:** `status`

---

**Local search.** In this paper, we consider four types of neighborhood operations applied to the current solution, i.e., adding, removing, and moving a CP (Lavielle, 1998) for a uniformly selected frequency $d$, and merging two adjacent CP locations uniformly sampled from $\tau$ at a random position between them. Schematic illustrations of these four operations are provided in Figures 2 and 3. We set the number of searching iterations at a specific temperature $c$ equal to the size of neighborhoods, i.e., $|\mathcal{D}^{\text{init}}| \cdot T$ (Aarts & Korst, 1989). Each operation is uniformly selected from adding, removing, and moving one CP. Subsequently, two adjacent CP locations are merged only once because this operation significantly fluctuates the objective function value. When the operation is not possible (e.g., removing operation for a frequency with no CP), it is skipped. If no transition to neighborhoods occurs when these operations are repeated sufficiently at a certain temperature $c$, the search is terminated.

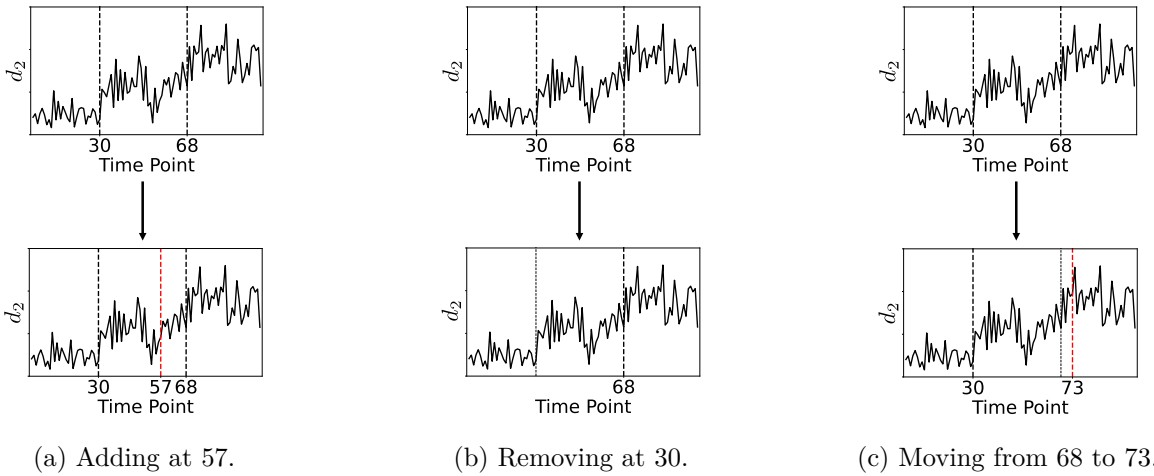

(a) Adding at 57.        (b) Removing at 30.        (c) Moving from 68 to 73.

Figure 2: Illustrations of the three local search operations for frequency $d$. (a) Adding is to insert a CP at a randomly selected time point with no CP. (b) Removing means to delete a CP at a randomly selected time point with a CP. (c) Moving is to shift a CP that is randomly selected from $\tau^{(d)}$ to a random position between its adjacent CPs.

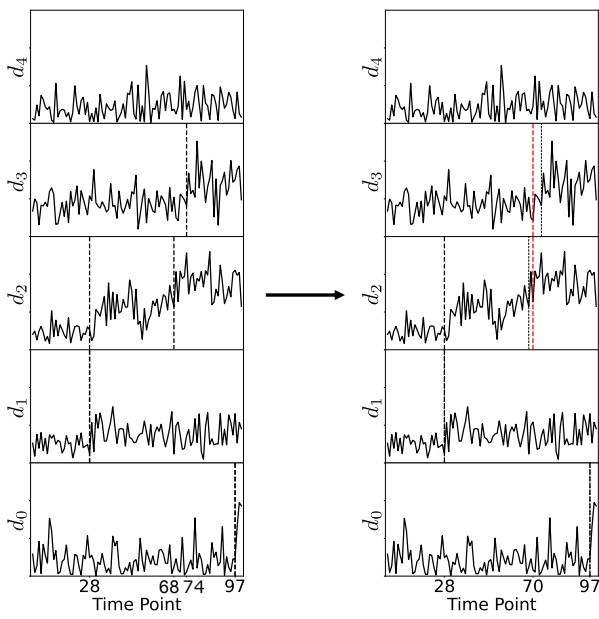

Figure 3: Illustration of the local search operation which merges two adjacent CP locations. In this figure, CP for frequency $d_2$ at time point 68 and CP for $d_3$ at 74 are merged at 70.

**Setting of initial temperature.** We determine an initial temperature $c_0$ by setting an acceptance ratio

$$\eta(c_0) = \frac{\text{\# accepted transitions}}{\text{\# proposed transitions}}$$

to a desired value in a preliminary experiment of simulated annealing. In practice, we start by setting the temperature to a sufficiently small positive value, then multiply it with a constant factor $\lambda^+$, larger than 1, as follows:

$$c_{i+1}^+ = \lambda^+ c_i^+,$$

where $c_i^+$ represents the $i$-th temperature in the initial value setting, until the acceptance ratio exceeds the predefined criterion (Aarts & Korst, 1989). Therefore, the initial temperature $c_0$ is specified as the final value of $c_i^+$.

**Decrement of temperature.** We employ a geometric cooling schedule, which is used as a practical method of temperature control, altough it does not guarantee the convergence to a global optimum. The decrement function of the $i$-th temperature $c_i$ is given as

$$c_{i+1} = \lambda c_i,$$

where $\lambda$ is a constant factor smaller than but close to 1. The value typically lies between 0.8 and 0.99 (Aarts & Korst, 1989). That is because the decreasing rate must be sufficiently slow to obtain a better solution for the optimization problem.

The overall procedure of the CP candidate selection using simulated annealing is shown in Algorithm 2.

---

**Algorithm 2** CP Candidate Selection by Simulated Annealing

---

**Input:** Time sequence $\boldsymbol{x}$
 1: Obtain spectral sequences $\boldsymbol{f}^{(d)}$ for $d \in \{0, \ldots, D-1\}$ in (3) by applying STFT to $\boldsymbol{x}$
 2: Initialize the CP candidates $\mathcal{T}$ as $\mathcal{T}^{\text{init}}$ in (7) using dynamic programming
 3: Determine the initial temperature $c$ in a preliminary experiment
 4: **while** true **do**
 5:     **while** The number of local search is less than $|\mathcal{D}^{\text{init}}| \cdot T$ **do**
 6:         Frequency $d$ is uniformly sampled from $\mathcal{D}^{\text{init}}$
 7:         The operation is uniformly sampled from adding, removing, and moving a CP for $d$
 8:         Obtain $\mathcal{T}'$ by applying the operation to $\mathcal{T}$
 9:         `status` $\leftarrow$ `metropolis_algorithm`$(\Delta E(\mathcal{T}', \mathcal{T}, \boldsymbol{x}), c)$
10:         **if** `status` is Acceptance **then**
11:            $\mathcal{T} \leftarrow \mathcal{T}'$
12:         **end if**
13:     **end while**
14:     Obtain $\mathcal{T}'$ by merging uniformly sampled two adjacent CP locations in $\mathcal{T}$
15:     `status` $\leftarrow$ `metropolis_algorithm`$(\Delta E(\mathcal{T}', \mathcal{T}, \boldsymbol{x}), c)$
16:     **if** `status` is Acceptance **then**
17:         $\mathcal{T} \leftarrow \mathcal{T}'$
18:     **end if**
19:     **if** no transition to neighborhoods occured at $c$ **then**
20:         **break**
21:     **end if**
22:     Update the temperature using a constant factor $\lambda$ as $c \leftarrow \lambda c$
23: **end while**
**Output:** CP candidates $\mathcal{T}$

---

# 4 Selective Inference on CP Candidate Location

In this section, using the SI framework, we quantify the statistical significance of all locations selected as CP candidates by the algorithm in Section 3 in the form of $p$-values. By setting the significance level $\alpha$ (e.g., 0.05 or 0.01) and considering CP candidate locations with the $p$-values below $\alpha$ as the final CPs, it is theoretically guaranteed that the false positive detection probabilities (type I error rates) of the final CPs are controlled below the specified significance level $\alpha$.

To formalize the SI framework, let us write the CP candidate selection algorithm in Section 3 as $\mathcal{A} \colon \boldsymbol{X} \mapsto \boldsymbol{\mathcal{T}}$ [5]. As detailed in Section 3, the output of this algorithm is a collection of CP candidates represented as $\boldsymbol{\mathcal{T}} = (\boldsymbol{\tau}^{(0)}, \ldots, \boldsymbol{\tau}^{(D-1)})$, where each $\boldsymbol{\tau}^{(d)} = \{\tau_1^{(d)}, \ldots, \tau_{K^{(d)}}^{(d)}\} \subseteq [T-1]$ is the ordered set of CP candidates for frequency $d \in \{0, \ldots, D-1\}$. The final set of selected CP candidate locations $\boldsymbol{\tau}$ is obtained by taking the union of these sets of CP candidates, i.e., $\boldsymbol{\tau} = \bigcup_{d=0}^{D-1} \boldsymbol{\tau}^{(d)} = \{\tau_1, \ldots, \tau_K\} \subseteq [T-1]$. Subsequently, we quantify the statistical significance of each CP candidate location $\tau_k \in \boldsymbol{\tau}$ for $k \in [K]$, in the form of $p$-values within the SI framework. Therefore, without loss of generality, we formulate the hypothesis testing problem for the $k$-th CP candidate location $\tau_k$ for simplicity.

## 4.1 Statistical Test on CP Candidate Location

**Hypotheses.** To test the statistical significance of CP candidate location $\tau_k$, we consider whether a mean-shift occurs before and after this location $\tau_k$ at any frequency for which this location was selected as a CP candidate. To formalize this, let $\mathcal{D}_k$ denote the set of frequencies where $\tau_k$ is selected as a CP candidate location, i.e., $\mathcal{D}_k = \{d \in \{0, \ldots, D-1\} \mid \tau_k \in \boldsymbol{\tau}^{(d)}\}$. For each $d \in \mathcal{D}_k$, let $\tau_{\mathrm{pre}}^{(d)}$ and $\tau_{\mathrm{suc}}^{(d)}$ represent the CP candidate locations immediately before and after $\tau_k$ at frequency $d$, respectively. Note that although $\tau_{\mathrm{pre}}^{(d)}$ and $\tau_{\mathrm{suc}}^{(d)}$ depend on $\tau_k$, this dependence is omitted from the notation for simplicity. Given these notations, we formulate the null hypothesis $\mathrm{H}_{0,k}$ and alternative hypothesis $\mathrm{H}_{1,k}$ as follows:

$$\mathrm{H}_{0,k} \colon \frac{1}{\tau_k - \tau_{\mathrm{pre}}^{(d)}} \sum_{t=\tau_{\mathrm{pre}}^{(d)}+1}^{\tau_k} \mu_t^{(d)} = \frac{1}{\tau_{\mathrm{suc}}^{(d)} - \tau_k} \sum_{t=\tau_k+1}^{\tau_{\mathrm{suc}}^{(d)}} \mu_t^{(d)}, \ \forall d \in \mathcal{D}_k, \tag{8}$$

$$\text{v.s.}$$

$$\mathrm{H}_{1,k} \colon \frac{1}{\tau_k - \tau_{\mathrm{pre}}^{(d)}} \sum_{t=\tau_{\mathrm{pre}}^{(d)}+1}^{\tau_k} \mu_t^{(d)} \neq \frac{1}{\tau_{\mathrm{suc}}^{(d)} - \tau_k} \sum_{t=\tau_k+1}^{\tau_{\mathrm{suc}}^{(d)}} \mu_t^{(d)}, \ \exists d \in \mathcal{D}_k. \tag{9}$$

**Test statistic.** We define the test statistic to test with the null hypothesis (8) and alternative hypothesis (9) as follows. First, for each frequency $d \in \mathcal{D}_k$, we consider the difference between the average values of the complex spectra in the two segments before and after the location $\tau_k$. Then, we aggregate these differences across all frequencies $d \in \mathcal{D}_k$ to define the test statistic. Therefore, the test statistic $T_k(\boldsymbol{X})$ is formulated as

$$T_k(\boldsymbol{X}) = \sigma^{-1} \sqrt{\sum_{d \in \mathcal{D}_k} \frac{a_{\mathrm{len}}^{(d)} c_{\mathrm{sym}}^{(d)}}{M} \left| \bar{F}_{\tau_{\mathrm{pre}}^{(d)}+1:\tau_k}^{(d)} - \bar{F}_{\tau_k+1:\tau_{\mathrm{suc}}^{(d)}}^{(d)} \right|^2}, \tag{10}$$

where $a_{\mathrm{len}}^{(d)} = (\tau_{\mathrm{suc}}^{(d)} - \tau_k)(\tau_k - \tau_{\mathrm{pre}}^{(d)})/(\tau_{\mathrm{suc}}^{(d)} - \tau_{\mathrm{pre}}^{(d)}) \in \mathbb{R}$ is a scaling factor for correcting the segment lengths before and after the location $\tau_k$ for each frequency $d \in \mathcal{D}_k$, $c_{\mathrm{sym}}^{(d)} \in \mathbb{R}$ defined in (6) is required for considering the complex conjugate symmetry of the spectrum, and $\bar{F}_{s:e}^{(d)} \in \mathbb{C}$ is the average value of the complex spectra in the segment from time points $s$ to $e$ for frequency $d \in \mathcal{D}_k$, i.e.,

$$\bar{F}_{s:e}^{(d)} = \frac{1}{e-s+1} \sum_{t=s}^{e} F_t^{(d)} = \frac{1}{e-s+1} \left( \mathbf{1}_{s:e} \otimes \boldsymbol{w}_M^{(d)} \right)^\top \boldsymbol{X}. \tag{11}$$

---

[5]Note that, to ensure deterministic behavior of the algorithm $\mathcal{A}$, the random seed is fixed to a constant value at the beginning of the procedure.

An important point when conducting the statistical test within the SI framework is that the test statistic is represented as the norm of a projection of the data, i.e., $T_k(\boldsymbol{X}) = \sigma^{-1}\|P_k\boldsymbol{X}\|$, where $P_k \in \mathbb{R}^{N \times N}$ is a projection matrix that depends on the observed time series $\boldsymbol{x}$ only through the selected CP candidates $\mathcal{A}(\boldsymbol{x})$ (see the next subsection for further details). We now define the orthogonal projection matrix $P_k$ as

$$P_k = \sum_{d \in \mathcal{D}_k} \frac{a_{\text{len}}^{(d)} c_{\text{sym}}^{(d)}}{M} \boldsymbol{v}^{(d)} \boldsymbol{v}^{(d)*\top} \in \mathbb{R}^{N \times N}, \tag{12}$$

where

$$\boldsymbol{v}^{(d)} = \frac{1}{\tau_k - \tau_{\text{pre}}^{(d)}} \left(\mathbf{1}_{\tau_{\text{pre}}^{(d)}+1:\tau_k} \otimes \boldsymbol{w}_M^{(d)}\right) - \frac{1}{\tau_{\text{suc}}^{(d)} - \tau_k} \left(\mathbf{1}_{\tau_k+1:\tau_{\text{suc}}^{(d)}} \otimes \boldsymbol{w}_M^{(d)}\right) \in \mathbb{C}^N \tag{13}$$

is a vector that depends on the selected CP candidates $\mathcal{A}(\boldsymbol{x})$, and $\boldsymbol{v}^{(d)*}$ is the complex conjugate of $\boldsymbol{v}^{(d)}$. Using (10)–(13), the test statistic $T_k(\boldsymbol{X})$ can be rewritten as

$$\begin{aligned} T_k(\boldsymbol{X}) &= \sigma^{-1}\sqrt{\sum_{d \in \mathcal{D}_k} \frac{a_{\text{len}}^{(d)} c_{\text{sym}}^{(d)}}{M} |\boldsymbol{v}^{(d)\top}\boldsymbol{X}|^2} \\ &= \sigma^{-1}\sqrt{\boldsymbol{X}^\top \left(\sum_{d \in \mathcal{D}_k} \frac{a_{\text{len}}^{(d)} c_{\text{sym}}^{(d)}}{M} \boldsymbol{v}^{(d)} \boldsymbol{v}^{(d)*\top}\right) \boldsymbol{X}} \\ &= \sigma^{-1}\sqrt{\boldsymbol{X}^\top P_k \boldsymbol{X}} = \sigma^{-1}\|P_k\boldsymbol{X}\| \quad (\because P_k = P_k^2, P_k = P_k^\top). \end{aligned} \tag{14}$$

## 4.2 Computing Selective $p$-value

**Selective Inference (SI).** To compute the $p$-value, we need to identify the sampling distribution of the test statistic $T_k(\boldsymbol{X})$. However, as the projection matrix $P_k$ within the test statistic $T_k(\boldsymbol{X})$ depends on the CP candidates $\mathcal{A}(\boldsymbol{X})$ (which, in turn, depends on the sequence $\boldsymbol{X}$ through the CP candidate selection algorithm $\mathcal{A}$), the sampling distribution of the test statistic $T_k(\boldsymbol{X})$ is too complicated to characterize. Within the SI framework, to address this challenge, we consider the sampling distribution of the test statistic conditional on the event that the selected CP candidates $\mathcal{A}(\boldsymbol{X})$ for a random sequence $\boldsymbol{X}$ is the same as $\mathcal{A}(\boldsymbol{x})$ for the observed sequence $\boldsymbol{x}$, that is,

$$T_k(\boldsymbol{X}) \mid \{\mathcal{A}(\boldsymbol{X}) = \mathcal{A}(\boldsymbol{x})\}. \tag{15}$$

To compute a selective $p$-value based on the conditional sampling distribution in (15), we introduce an additional condition on the sufficient statistic of the nuisance parameter $\mathcal{Q}(\boldsymbol{X})$, which is defined as

$$\mathcal{Q}(\boldsymbol{X}) = (\mathcal{V}(\boldsymbol{X}), \mathcal{U}(\boldsymbol{X})) \in \mathbb{R}^N \times \mathbb{R}^N, \tag{16}$$

with

$$\mathcal{V}(\boldsymbol{X}) = \frac{\sigma P_k \boldsymbol{X}}{\|P_k\boldsymbol{X}\|} \in \mathbb{R}^N, \quad \mathcal{U}(\boldsymbol{X}) = (I_N - P_k)\boldsymbol{X} \in \mathbb{R}^N.$$

This additional conditioning on $\mathcal{Q}(\boldsymbol{X})$ is a standard approach for computational tractability in the SI literature (Loftus & Taylor, 2015). Based on the additional conditioning on $\mathcal{Q}(\boldsymbol{X})$, the following theorem tells that the conditional distribution of the test statistic can be represented as a truncated $\chi$-distribution.

**Theorem 1.** *Consider a random sequence $\boldsymbol{X} \sim \mathcal{N}(\boldsymbol{s}, \sigma^2 I_N)$ and an observed sequence $\boldsymbol{x}$. Let $\mathcal{A}(\boldsymbol{X})$ and $\mathcal{A}(\boldsymbol{x})$ be the detected CP candidates, by applying a CP candidate selection algorithm to $\boldsymbol{X}$ and $\boldsymbol{x}$, respectively. Let $P_k \in \mathbb{R}^{N \times N}$ be a projection matrix depending on $\mathcal{A}(\boldsymbol{x})$, and consider a test statistic in the form of $T_k(\boldsymbol{X}) = \sigma^{-1}\|P_k\boldsymbol{X}\|$. Furthermore, define the nuisance parameter $\mathcal{Q}(\boldsymbol{X}) = (\mathcal{V}(\boldsymbol{X}), \mathcal{U}(\boldsymbol{X}))$ as in (16).*

*Then, under the null hypothesis that the norm of the true signal vector $\boldsymbol{s}$ projected by $P_k$ is zero (i.e., $\|P_k\boldsymbol{s}\| = 0$), the conditional distribution of the test statistic*

$$T_k(\boldsymbol{X}) \mid \{\mathcal{A}(\boldsymbol{X}) = \mathcal{A}(\boldsymbol{x}), \mathcal{Q}(\boldsymbol{X}) = \mathcal{Q}(\boldsymbol{x})\}$$

is a truncated $\chi$-distribution $\mathrm{TC}(\mathrm{tr}(P_k), \mathcal{Z})$ with the degrees of freedom $\mathrm{tr}(P_k)$ and the truncation region $\mathcal{Z}$. The truncation region $\mathcal{Z}$ is defined as

$$\mathcal{Z} = \{z \in \mathbb{R} \mid \mathcal{A}(\boldsymbol{a} + \boldsymbol{b}z) = \mathcal{A}(\boldsymbol{x})\},\ \boldsymbol{a} = \mathcal{U}(\boldsymbol{x}),\ \boldsymbol{b} = \mathcal{V}(\boldsymbol{x}).$$

The proof of Theorem 1 is deferred to Appendix B.1. By using the sampling distribution of the test statistic $T_k(\boldsymbol{X})$ conditional on $\mathcal{A}(\boldsymbol{X}) = \mathcal{A}(\boldsymbol{x})$ and $\mathcal{Q}(\boldsymbol{X}) = \mathcal{Q}(\boldsymbol{x})$ in Theorem 1, we can define the selective $p$-value as

$$p_k^{\text{selective}} = \mathbb{P}_{\mathrm{H}_{0,k}}\left(T_k(\boldsymbol{X}) \geq T_k(\boldsymbol{x}) \mid \mathcal{A}(\boldsymbol{X}) = \mathcal{A}(\boldsymbol{x}), \mathcal{Q}(\boldsymbol{X}) = \mathcal{Q}(\boldsymbol{x})\right). \tag{17}$$

**Theorem 2.** *The selective $p$-value defined in (17) satisfies the property:*

$$\mathbb{P}_{\mathrm{H}_{0,k}}\left(p_k^{\text{selective}} \leq \alpha \mid \mathcal{A}(\boldsymbol{X}) = \mathcal{A}(\boldsymbol{x})\right) = \alpha,\ \forall \alpha \in (0, 1).$$

*Then, the selective $p$-value also satisfies the following property of a valid $p$-value:*

$$\mathbb{P}_{\mathrm{H}_{0,k}}(p_k^{\text{selective}} \leq \alpha) = \alpha,\ \forall \alpha \in (0, 1).$$

The proof of Theorem 2 is deferred to Appendix B.2. This theorem guarantees that the selective $p$-value is uniformly distributed under the null hypothesis $\mathrm{H}_{0,k}$, and thus can be used to conduct the valid statistical inference in (8) and (9). Figure 4 schematically illustrates the SI framework presented in this section.

Once the truncation region $\mathcal{Z}$ is identified, the selective $p$-value in (17) can be easily computed by Theorem 1. Thus, the remaining task is reduced to identifying the truncation region $\mathcal{Z}$. In this study, to identify the truncation region $\mathcal{Z}$, we adopt the method based on parametric-programming technique proposed by Duy & Takeuchi (2022). For further details, refer to Appendix C.

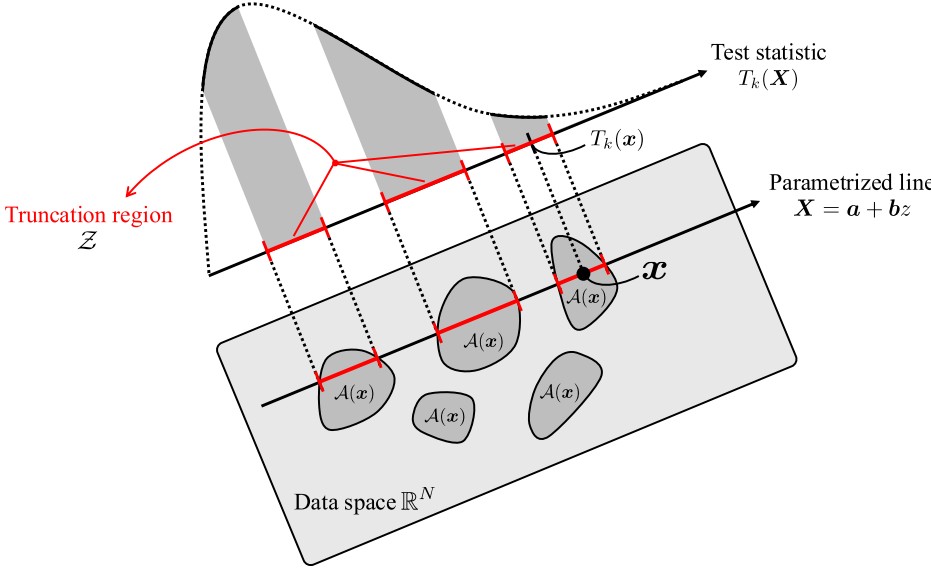

Figure 4: Schematic illustration of the SI framework. A point in the data space $\mathbb{R}^N$ corresponds to a sequence with length $N$. The darkly shaded regions in the data space indicate that, if we input a point in these regions into the algorithm $\mathcal{A}$, the CP candidates are the same as $\mathcal{A}(\boldsymbol{x})$ obtained from the observed sequence $\boldsymbol{x}$. By conditioning on these regions and $\mathcal{Q}(\boldsymbol{X}) = \mathcal{Q}(\boldsymbol{x})$, the conditional sampling distribution of the test statistic $T_k(\boldsymbol{X})$ is represented as a truncated $\chi$-distribution. Thus, selective $p$-values are defined based on the tail probability of such a truncated $\chi$-distribution.

## 5 Numerical Experiments

### 5.1 Methods for Comparison

In our experiments, we compared the proposed method (`Proposed`) using $p_k^{\text{selective}}$ in (17) with the following methods in terms of type I error rate control and power.

- `OC`: In this method, that is, a simple extension of SI literature to our setting, we consider $p$-values with additional conditioning (over-conditioning) described in Appendix C.1.

- `OptSeg-SI-oc`, `OptSeg-SI` (Duy et al., 2020): These methods use $p$-values conditioned only on the dynamic programming algorithm, disregarding the conditioning on simulated annealing.

- `Naive`: This method is a conventional statistical inference.

- `Bonferroni`: This method applies Bonferroni correction for multiple testing correction.

The details of these comparison methods are provided in Appendix D.1.

### 5.2 Synthetic Data Experiments

**Experimental setup.** In all synthetic experiments, we set window size $M \in \{512, 1024\}$, the number of frequencies $D = \lfloor \frac{M}{2} \rfloor + 1$, the length of sequence $N = M \cdot T$, where $T$ was specified for each experiment, the sampling rate $f_s = 20480$, and each element of mean vector $\boldsymbol{s}$ as

$$s_n = \sum_{d \in \{d_1, d_2, d_3\}} A_n^{(d)} \sin\left(\omega^{(d)}(n-1)\right) \quad (1 \le n \le N),$$

where frequencies $d_1, d_2, d_3 \in \{0, ..., D-1\}$ were randomly selected without replacement for each simulation, $A_n^{(d)}$ was defined for each experiment, and $\omega^{(d)} \in \left\{ 2\pi(\frac{f_s}{M})d \,\middle|\, d = 0, \ldots, \lfloor \frac{M}{2} \rfloor \right\}$. We used BIC for the choice of penalty parameters $\boldsymbol{\beta}$ and $\gamma$ as indicated in Appendix A, and set the parameters of simulated annealing as $c_0^+ = 1000, \lambda^+ = 1.5, \eta(c_0) = 0.5$ and $\lambda = 0.8$ in Section 3.4. After detecting CP candidates, a CP candidate location $\tau_k^{\text{det}}$ randomly selected from $\boldsymbol{\tau}^{\text{det}}$ was tested at the significance level $\alpha = 0.05$.

In the experiments conducted to evaluate the type I error rate control, we generated 1000 null sequences, which did not contain true CPs in the frequency domain, $\boldsymbol{x} = (x_1, \ldots, x_N)^\top \sim \mathcal{N}\left(\boldsymbol{s}, \sigma^2 I_N\right)$, where $A_n^{(d)} = A^{(d)}$ was randomly sampled from $[0, 1)$ for $d$ in each simulation, and $\sigma = 1$, for each $T \in \{40, 60, 80, 100\}$.

Regarding the experiments to compare the power, we generated sequences $\boldsymbol{x} = (x_1, \ldots, x_N)^\top \sim \mathcal{N}\left(\boldsymbol{s}, \sigma^2 I_N\right)$, where

$$A_n^{(d)} = \begin{cases} A^{(d)} & \text{if } 1 \le t \le M \cdot t_1^{(d)}, \\ A^{(d)} + \Delta & \text{if } M \cdot t_1^{(d)} + 1 \le t \le M \cdot t_2^{(d)}, \\ A^{(d)} + 2\Delta & \text{if } M \cdot t_2^{(d)} + 1 \le t \le T, \end{cases}$$

with $A^{(d_1)}, A^{(d_2)}, A^{(d_3)} \in [0, 1)$ which were randomly sampled in each simulation, $\left(t_1^{(d_1)}, t_1^{(d_2)}, t_1^{(d_3)}\right) = (18, 20, 22)$, $\left(t_2^{(d_1)}, t_2^{(d_2)}, t_2^{(d_3)}\right) = (38, 40, 42)$, an intensity of the change $\Delta \in \{0.04, 0.08, 0.12, 0.16\}$ and $\sigma = 1$, for $T = 60$. In each case, we ran 1000 trials. Since we tested only when a CP candidate location was correctly detected, the power was defined as follows:

$$\text{Power (or Conditional Power)} = \frac{\text{\# correctly detected \& rejected}}{\text{\# correctly detected}}.$$

We considered the CP candidate location $\tau_k$ to be correctly detected if it satisfied the following two conditions:

- The set $\mathcal{D}_k$ of frequencies containing at least one CP candidate was a subset of $\{d_1, d_2, d_3\}$.

- For $\mathcal{D}_k$ satisfying the above condition, either $\min_{d \in \mathcal{D}_k} t_1^{(d)} \le \tau_k \le \max_{d \in \mathcal{D}_k} t_1^{(d)}$ or $\min_{d \in \mathcal{D}_k} t_2^{(d)} \le \tau_k \le \max_{d \in \mathcal{D}_k} t_2^{(d)}$ held, that is, $\tau_k$ was detected within the true CP locations for the frequencies in $\mathcal{D}_k$.

**Experimental results.** The results of experiments regarding the control of the type I error rate are shown in Figure 5. The `Proposed`, `OC`, and `Bonferroni` successfully controlled the type I error rate below the significance level, whereas the `OptSeg-SI`, `OptSeg-SI-oc`, and `Naive` could not. That was because the `OptSeg-SI` and `OptSeg-SI-oc` used $p$-values conditioned only on the dynamic programming algorithm, excluding the conditioning on simulated annealing, and the `Naive` employed conventional $p$-values without conditioning. Since the `OptSeg-SI`, `OptSeg-SI-oc`, and `Naive` failed to control the type I error rate, we omitted the analysis of their power. The results of power experiments are shown in Figure 6. Based on these results, the `Proposed` was the most powerful of all methods that controlled the type I error rate. The power of the `OC` was lower than that of the `Proposed` due to redundant conditions (see Appendix C for details). Furthermore, the `Bonferroni` method had the lowest power because it was a highly conservative approach that accounted for the huge number of all possible hypotheses. Additionally, we provide the computational time of the `Proposed` in both experiments and the information on the computer resources in Appendix D.2.

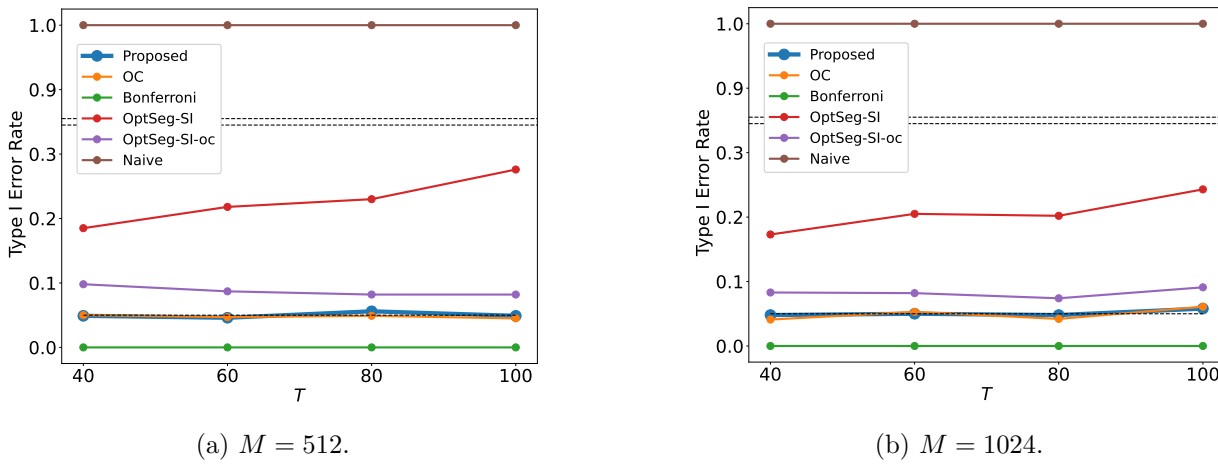

(a) $M = 512$.

(b) $M = 1024$.

Figure 5: Type I error rate.

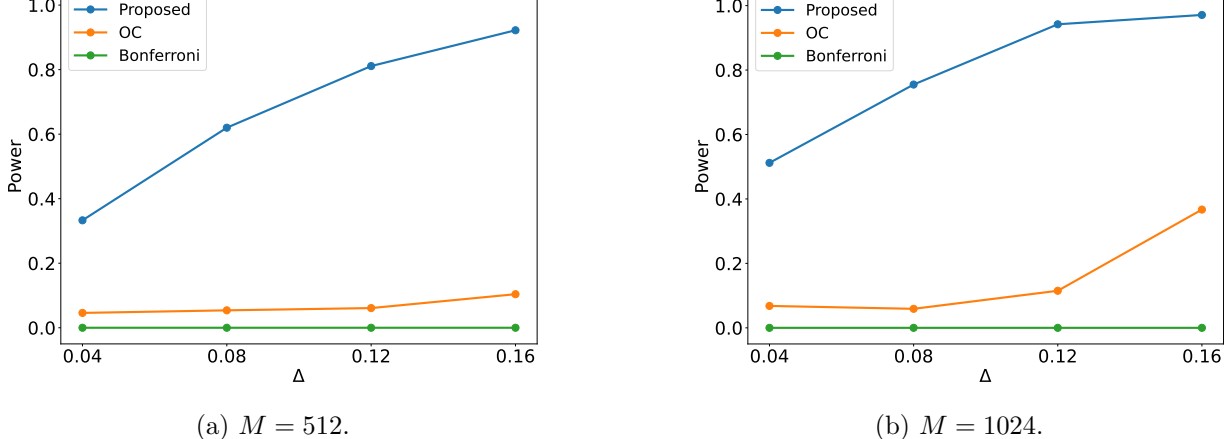

(a) $M = 512$.

(b) $M = 1024$.

Figure 6: Power.

**Additional experiments.** We also conducted the following experiments to investigate the robustness of the `Proposed` in terms of type I error rate control.

- Unknown noise variance: We considered the case where the variance $\sigma^2$ was estimated from the same data.

- Non-Gaussian noise: We also considered the case where the noise followed the five types of standardized non-Gaussian distributions.

- Correlated noise: Furthermore, we considered the sequence whose noise was correlated, i.e., the covariance matrix $\Sigma \neq \sigma^2 I_N$. In this case, although the test statistic did not theoretically follow a $\chi$-distribution, we conducted the hypothesis testing using our proposed framework.

In addition, a sensitivity study for the penalty parameter $\gamma$ within the objective function in (5) was also performed. These details and results are shown in Appendix D.3 and D.4, respectively.

## 5.3 Real Data Experiments

To demonstrate the practical applicability of the `Proposed`, we applied the `Proposed`, `OC`, and `Naive` to a real-world dataset. We used the set No.2 of the IMS bearing dataset, which is provided by the Center for Intelligent Maintenance Systems (IMS), University of Cincinnati (Qiu et al., 2006) and is available from the Prognostic data repository of NASA (Lee et al., 2007). The experimental apparatus consisted of four identical bearings (bearings 1, 2, 3, and 4) installed on a common shaft, driven at a constant rotation speed by an AC motor under applied radial loading. In this dataset, the vibration signals were measured using accelerometers until the outer race of bearing 1 failed at the end of the experiment, as shown in Figure 7. The signal with the sampling rate $f_s = 20480$ was recorded for one second at intervals of 10 minutes over a period of about 7 days for each bearing. The analysis for the signal of bearing 1 in the frequency domain, as conducted by Gousseau et al. (2016), had revealed that enhancements of the spectral intensity were detected in harmonics of the characteristic frequency (236 Hz) associated with the outer race failure (Ball Pass Frequency Outer race, BPFO) on 3–4 days. Based on this previous study, we conducted CP candidate selection for the sensor data of bearing 1 in the frequency domain for two periods: 0.25–2.25 days when no significant changes in the spectra existed and 2.25–4.25 days when the BPFO harmonics exhibited the spectral amplification. For each period, we computed the DFT of $M = 1024$ consecutive points in the 20480 samples and repeated the procedure $T = 2$ days/10 min $= 288$ times. As a result of the DFT, we obtained components for 513 frequencies (0, 20, ..., 10240 Hz) and conducted CP candidate selection using 131 frequencies within the range of 1400–4000 Hz, which included the 6th to 16th harmonics of the characteristic frequency. For other parameters related to the CP candidate selection algorithm, please refer to the experimental setup in Section 5.2. Subsequently, we tested the detected CP candidate locations to evaluate whether each of them was a genuine CP location. The variance $\sigma^2$ for testing was estimated from the data on 0–0.25 days that did not contain significant spectral changes and was not used in any of the experiments. The results for the signal of bearing 1 on 0.25–2.25 days and 2.25–4.25 days are shown in Figure 8. In panel (a), the time variation of a frequency spectrum (1920 Hz) where a CP candidate location was falsely detected is shown for the period of 0.25–2.25 days. It shows that $p$-values of the `Proposed` and `OC` are above the significance level 0.05, and therefore the result provides the validity of the inference, while $p$-value of the `Naive` is too small. In panel (b), the time variations of the 8th and 15th harmonics of the BPFO where CP candidate locations were correctly detected are presented for the period of 2.25–4.25 days. In this case, $p$-values of the `Proposed` are below the significance level $\frac{0.05}{2} = 0.025$ decided by Bonferroni correction, thus it indicates that the inference is valid. In contrast, $p$-values of the `OC` are too large, due to the loss of power caused by the redundant conditions. In addition, since even the time sequences of the healthy bearings 2, 3, and 4 had been reported to indicate spectral amplification associated with the outer race fault in bearing 1 (Gousseau et al., 2016), we performed the same analysis for the three signals. The results are shown in Appendix D.5.

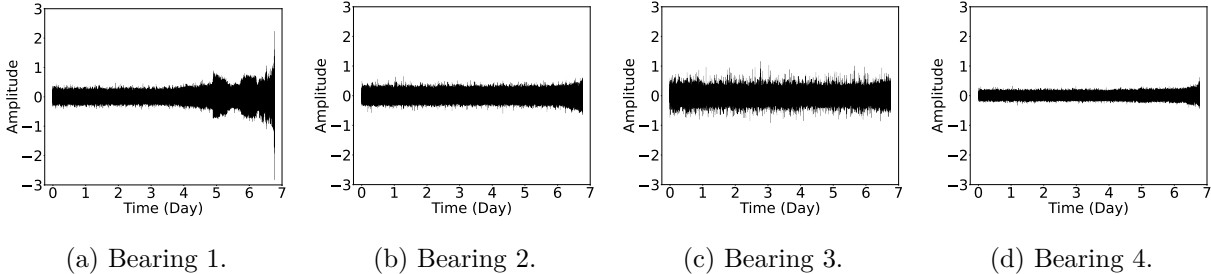

(a) Bearing 1.     (b) Bearing 2.     (c) Bearing 3.     (d) Bearing 4.

Figure 7: Vibration signals of the four bearings in the set No.2 of the IMS bearing dataset. Each signal was measured using an accelerometer for about 7 days until the outer race of bearing 1 failed. We used not only the signal of the damaged bearing 1 but also those of the other healthy bearings for the analysis because the spectral amplification associated with the failure of bearing 1 had been reported to be observed in the signals of all four bearings (Gousseau et al., 2016).

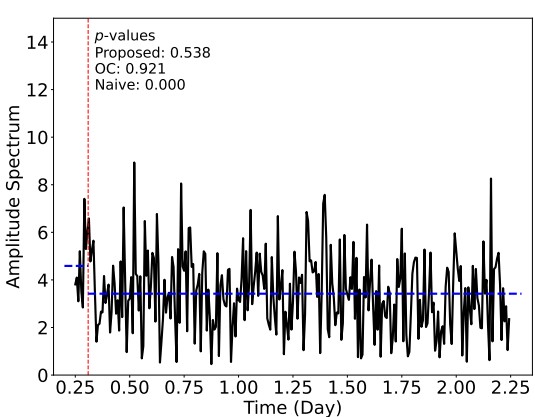

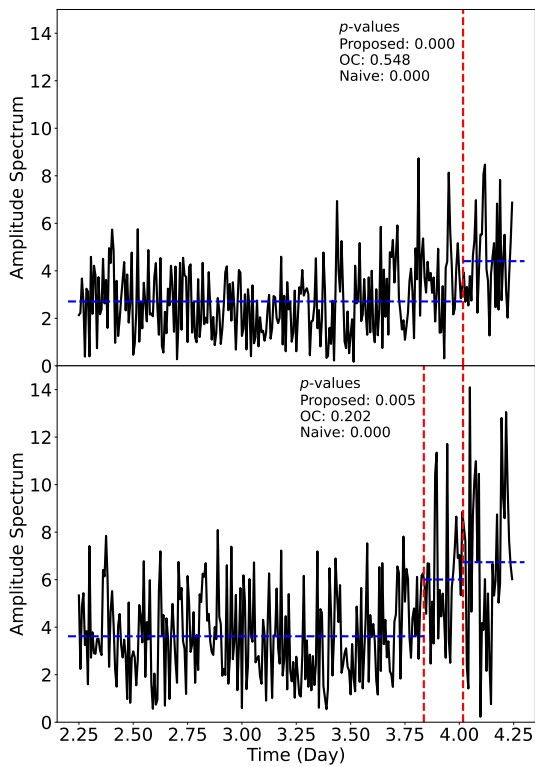

(a) Inference on a falsely detected CP candidate location for 1920 Hz (around the 8th harmonic) on 0.25–2.25 days.

(b) Inferences on truely detected CP candidate locations for 1880 Hz (the 8th harmonic, above) and 3540 Hz (the 15th harmonic, below) on 2.25–4.25 days.

Figure 8: Results of the CP candidate selection for the signal of bearing 1 in the frequency domain and the subsequent inference for the detected CP candidate locations. In panel (b), note that $p$-values for the first CP candidate location were actually computed by considering not only a CP candidate of the 15th harmonic but also a CP candidate of 1900 Hz (around the 8th harmonic).

# 6 Conclusion, Limitation, and Future Work

In this paper, we developed a statistical inference method to quantify the statistical significance of detected CP locations in the frequency domain. Our proposed framework contributes to various fields where time-frequency analysis is widely employed, such as condition monitoring of machine systems using vibrational, electrical, and acoustic signals, and medical diagnosis based on biosignals. We conducted comprehensive experiments on both synthetic and real-world datasets. The results theoretically confirmed that our method provided an unbiased evaluation based on SI framework and demonstrated its superior performance compared to existing methods. However, an important limitation is the assumption that the noise follows an uncorrelated normal distribution, which is technically essential for deriving the conditional sampling distribution of the test statistic as a truncated $\chi$-distribution. In future work, we will extend our method to the case of multi-dimensional sequences. For instance, analyzing different types of time series obtained from multiple sensors would reveal significant changes in the system unattainable through the univariate approach, and reliability guarantee for the detections also provide a valuable future contribution.

### Acknowledgments

This work was partially supported by JST CREST (JPMJCR21D3, JPMJCR22N2), JST Moonshot R&D (JPMJMS2033-05), and RIKEN Center for Advanced Intelligence Project.

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

## A  Determination of Penalty Parameters in the Optimization Problem

In this section, we derive the penalty term in (5) used for detecting CP candidates in the frequency domain. We begin by assuming that the true mean vector of $\boldsymbol{f}^{(d)}$ for frequency $d \in \{0, \ldots, D-1\}$ is piecewise constant as follows:

$$f_t^{(d)} \sim \mathcal{CN}\left(\mu_{\text{seg}}^{(d)}\left(\tau_{k-1}^{(d)} : \tau_k^{(d)}\right), \sigma_f^2\right), t \in \left\{\tau_{k-1}^{(d)} + 1, \ldots, \tau_k^{(d)}\right\}, k \in [K^{(d)} + 1], \tag{18}$$

where $\mu_{\text{seg}}^{(d)}\left(\tau_{k-1}^{(d)} : \tau_k^{(d)}\right)$ represents the true mean of the $k$-th segment in frequency $d$, and $\sigma_f^2$ denotes the known variance such that $\sigma_f^2 = M\sigma^2$ by the property of DFT[6]. Then, we introduce BIC for deriving the

---

[6]Although we impose the assumption of the piecewise constant mean structure for model selection based on BIC, the theoretical validity of $p$-values obtained by our proposed SI method is guaranteed even when this assumption does not hold.

values of the penalty parameters $\boldsymbol{\beta}$ in (5). Given the assumptions of (18), the unknown parameters $\boldsymbol{\theta}$ of the sequence $\boldsymbol{f}^{(d)}$ are expressed as

$$\boldsymbol{\theta} = \left( \tau_1^{(d)}, \ldots, \tau_{K^{(d)}}^{(d)}, \mu_{\text{seg}}^{(d)} \left( \tau_0^{(d)} : \tau_1^{(d)} \right), \ldots, \mu_{\text{seg}}^{(d)} \left( \tau_{K^{(d)}}^{(d)} : \tau_{K^{(d)}+1}^{(d)} \right) \right).$$

Therefore, the degrees of freedom can be computed as $K^{(d)} + c_{\text{sym}}^{(d)} \left( K^{(d)} + 1 \right)$. That is because the mean vector consists of real numbers for frequency $d = 0, \frac{M}{2}$, and complex numbers for the other frequencies. Consequently, the BIC of this model is given by

$$\text{BIC} = -2 \log L(\boldsymbol{\theta}) + \left( K^{(d)} + c_{\text{sym}}^{(d)} \left( K^{(d)} + 1 \right) \right) \log T,$$

where $L$ denotes the likelihood function for this model, and it can be rewritten by ignoring the terms that do not contribute to its minimization as

$$\text{BIC} = \sum_{k=1}^{K^{(d)}+1} \mathcal{C} \left( \boldsymbol{f}_{\tau_{k-1}^{(d)}+1:\tau_k^{(d)}}^{(d)} \right) + \left( \left( c_{\text{sym}}^{(d)} + 1 \right) M\sigma^2 \log T \right) K^{(d)}.$$

Comparing the BIC with (7), $\beta^{(d)}$ can be defined as

$$\beta^{(d)} = \left( c_{\text{sym}}^{(d)} + 1 \right) M\sigma^2 \log T. \tag{19}$$

While $\boldsymbol{\beta}$ can be determined based on the BIC, to the best of our knowledge, there is no theoretical method to define $\gamma$. Therefore, referring to (19), we employ the value scaled by $M\sigma^2 \log T$ as $\gamma$, which is given by

$$\gamma = \kappa M\sigma^2 \log T, \tag{20}$$

where $\kappa$ is a hyper-parameter determined by users based on problem settings. We heuristically set $\kappa = 0.5$ in the synthetic data experiments, and $\kappa = 3$ in the real data experiment.

## B  Proofs

### B.1  Proof of Theorem 1

**Proof.**  According to the condition on $\mathcal{Q}(\boldsymbol{X}) = \mathcal{Q}(\boldsymbol{x})$, i.e., $\mathcal{U}(\boldsymbol{X}) = \mathcal{U}(\boldsymbol{x})$ and $\mathcal{V}(\boldsymbol{X}) = \mathcal{V}(\boldsymbol{x})$, we have

$$\begin{aligned} \mathcal{U}(\boldsymbol{X}) &= \mathcal{U}(\boldsymbol{x}) \\ \Leftrightarrow (I_N - P_k)\boldsymbol{X} &= \mathcal{U}(\boldsymbol{x}) \\ \Leftrightarrow \boldsymbol{X} &= \mathcal{U}(\boldsymbol{x}) + \mathcal{V}(\boldsymbol{X})z \\ \Leftrightarrow \boldsymbol{X} &= \mathcal{U}(\boldsymbol{x}) + \mathcal{V}(\boldsymbol{x})z \quad (\because \mathcal{V}(\boldsymbol{X}) = \mathcal{V}(\boldsymbol{x})) \\ \Leftrightarrow \boldsymbol{X} &= \boldsymbol{a} + \boldsymbol{b}z, \end{aligned}$$

where $\boldsymbol{a} = \mathcal{U}(\boldsymbol{x}), \boldsymbol{b} = \mathcal{V}(\boldsymbol{x})$, and $z = T_k(\boldsymbol{X}) = \sigma^{-1}\|P_k\boldsymbol{X}\|$. Then, we have

$$\begin{aligned} &\{ \boldsymbol{X} \in \mathbb{R}^N \,|\, \mathcal{A}(\boldsymbol{X}) = \mathcal{A}(\boldsymbol{x}), \mathcal{Q}(\boldsymbol{X}) = \mathcal{Q}(\boldsymbol{x}) \} \\ =&\{ \boldsymbol{X} \in \mathbb{R}^N \,|\, \mathcal{A}(\boldsymbol{X}) = \mathcal{A}(\boldsymbol{x}), \boldsymbol{X} = \boldsymbol{a} + \boldsymbol{b}z, z \in \mathbb{R} \} \\ =&\{ \boldsymbol{X} = \boldsymbol{a} + \boldsymbol{b}z \in \mathbb{R}^N \,|\, \mathcal{A}(\boldsymbol{a} + \boldsymbol{b}z) = \mathcal{A}(\boldsymbol{x}), z \in \mathbb{R} \} \\ =&\{ \boldsymbol{X} = \boldsymbol{a} + \boldsymbol{b}z \in \mathbb{R}^N \,|\, z \in \mathcal{Z} \}. \end{aligned}$$

Therefore, by noting that $\|P_k\boldsymbol{s}\|$ is zero, we obtain

$$T(\boldsymbol{X}) \,|\, \{ \mathcal{A}(\boldsymbol{X}) = \mathcal{A}(\boldsymbol{x}), \mathcal{Q}(\boldsymbol{X}) = \mathcal{Q}(\boldsymbol{x}) \} \sim \text{TC}(\text{tr}(P_k), \mathcal{Z}).$$

## B.2 Proof of Theorem 2

**Proof.** The sampling distribution of the test statistic conditional on $\mathcal{A}(\boldsymbol{X}) = \mathcal{A}(\boldsymbol{x})$ and $\mathcal{Q}(\boldsymbol{X}) = \mathcal{Q}(\boldsymbol{x})$ denoted by

$$T_k(\boldsymbol{X}) \,|\, \{\mathcal{A}(\boldsymbol{X}) = \mathcal{A}(\boldsymbol{x}), \mathcal{Q}(\boldsymbol{X}) = \mathcal{Q}(\boldsymbol{x})\}$$

is a truncated $\chi$-distribution with the degrees of freedom $\mathrm{tr}(P_k)$ and the truncation region $\mathcal{Z}$ defined in Theorem 1. Thus, by applying the probability integral transform, under the null hypothesis,

$$p_k^{\mathrm{selective}} \,|\, \{\mathcal{A}(\boldsymbol{X}) = \mathcal{A}(\boldsymbol{x}), \mathcal{Q}(\boldsymbol{X}) = \mathcal{Q}(\boldsymbol{x})\} \sim \mathrm{Unif}(0,1),$$

which leads to

$$\mathbb{P}_{\mathrm{H}_0,\mathrm{k}} \left( p_k^{\mathrm{selective}} \leq \alpha \,|\, \mathcal{A}(\boldsymbol{X}) = \mathcal{A}(\boldsymbol{x}), \mathcal{Q}(\boldsymbol{X}) = \mathcal{Q}(\boldsymbol{x}) \right) = \alpha, \; \forall \alpha \in (0,1).$$

Next, for any $\alpha \in (0,1)$, we have

$$\mathbb{P}_{\mathrm{H}_0,\mathrm{k}} \left( p_k^{\mathrm{selective}} \leq \alpha \,|\, \mathcal{A}(\boldsymbol{X}) = \mathcal{A}(\boldsymbol{x}) \right)$$

$$= \int \mathbb{P}_{\mathrm{H}_0,\mathrm{k}} \left( p_k^{\mathrm{selective}} \leq \alpha \,|\, \mathcal{A}(\boldsymbol{X}) = \mathcal{A}(\boldsymbol{x}), \mathcal{Q}(\boldsymbol{X}) = \mathcal{Q}(\boldsymbol{x}) \right) \mathbb{P}_{\mathrm{H}_0,\mathrm{k}} \left( \mathcal{Q}(\boldsymbol{X}) = \mathcal{Q}(\boldsymbol{x}) \,|\, \mathcal{A}(\boldsymbol{X}) = \mathcal{A}(\boldsymbol{x}) \right) d\mathcal{Q}(\boldsymbol{x})$$

$$= \alpha \int \mathbb{P}_{\mathrm{H}_0,\mathrm{k}}(\mathcal{Q}(\boldsymbol{X}) = \mathcal{Q}(\boldsymbol{x}) \,|\, \mathcal{A}(\boldsymbol{X}) = \mathcal{A}(\boldsymbol{x})) d\mathcal{Q}(\boldsymbol{x})$$

$$= \alpha.$$

Therefore, we obtain the result in Theorem 2 as follows:

$$\mathbb{P}_{\mathrm{H}_0,\mathrm{k}} \left( p_k^{\mathrm{selective}} \leq \alpha \right) = \sum_{\mathcal{A}(\boldsymbol{x})} \mathbb{P}_{\mathrm{H}_0,\mathrm{k}} \left( p_k^{\mathrm{selective}} \leq \alpha \,|\, \mathcal{A}(\boldsymbol{X}) = \mathcal{A}(\boldsymbol{x}) \right) \mathbb{P}_{\mathrm{H}_0,\mathrm{k}}(\mathcal{A}(\boldsymbol{X}) = \mathcal{A}(\boldsymbol{x}))$$

$$= \alpha \sum_{\mathcal{A}(\boldsymbol{x})} \mathbb{P}_{\mathrm{H}_0,\mathrm{k}}(\mathcal{A}(\boldsymbol{X}) = \mathcal{A}(\boldsymbol{x}))$$

$$= \alpha.$$

# C Identification of Truncation Region

In general, it is difficult to identify the truncation region $\mathcal{Z}$ in Theorem 1 directly because conditioning only on the result of the CP candidate selection, i.e., $\mathcal{A}(\boldsymbol{a} + \boldsymbol{b}z) = \mathcal{A}(\boldsymbol{x})$, is intractable. In this case, we have to enumerate all patterns where $\mathcal{A}(\boldsymbol{x})$ appears as a result of simulated annealing algorithm, which is computationally impractical. To address this issue, we first compute the region conditioned on the process of the algorithm $\mathcal{A}$. That is to say, it is guaranteed that the process remains identical within the region. This additional conditioning is often denoted as "over-conditioning" because it is redundant for valid inference. In the case of SI with over-conditioning, the type I error rate can still be controlled at the significance level, while the power tends to be low (Lee et al., 2016; Liu et al., 2018; Duy & Takeuchi, 2022). Therefore, we apply an efficient line search method based on parametric programming to compute $\mathcal{Z}$ where the redundant conditioning is removed for the purpose of improving the power (Duy & Takeuchi, 2022). In the following, we first compute the over-conditioned region in Appendix C.1, and then identify the truncation region $\mathcal{Z}$ using parametric programming in Appendix C.2.

## C.1 Over-Conditioning

Since we only need to consider one-dimensional data space in $\mathbb{R}^N$, we define the over-conditioned region $\mathcal{Z}^{\mathrm{oc}}$ where the process of the algorithm $\mathcal{A}$ remains unchanged as

$$\mathcal{Z}^{\mathrm{oc}}(\boldsymbol{a} + \boldsymbol{b}z) = \{r \in \mathbb{R} \,|\, \mathcal{S}_{\mathrm{DP}}(r) = \mathcal{S}_{\mathrm{DP}}(z), \mathcal{S}_{\mathrm{pre\text{-}SA}}(r) = \mathcal{S}_{\mathrm{pre\text{-}SA}}(z), \mathcal{S}_{\mathrm{SA}}(r) = \mathcal{S}_{\mathrm{SA}}(z)\}, \tag{21}$$

where $\mathcal{S}_{\mathrm{DP}}$, $\mathcal{S}_{\mathrm{pre\text{-}SA}}$ and $\mathcal{S}_{\mathrm{SA}}$ are the events characterized by the process of dynamic programming for generating an initial solution, the preliminary experiment to determine the initial temperature, and simulated annealing for making the solution more sophisticated, respectively. This conditioning is redundant because $\mathcal{Z}^{\mathrm{oc}}(\boldsymbol{a} + \boldsymbol{b}z)$ is a subset of the minimum conditioned region $\mathcal{Z}$.

**Over-conditioning on dynamic programming.** We compute $\{r \in \mathbb{R} \,|\, \mathcal{S}_{\mathrm{DP}}(r) = \mathcal{S}_{\mathrm{DP}}(z)\}$ by conditioning on all the operations based on the Bellman equation which is used in the dynamic programming algorithm to obtain the optimal solution in (7). Since the Bellman equation consists of the cost $\mathcal{C}(\cdot)$ and the penalty $\beta$, the condition is finally represented by a quadratic inequality, as in the following discussion of simulated annealing. The detail derivation is presented in Duy et al. (2020).

**Over-conditioning on simulated annealing containing the preliminary experiment.** Considering the algorithm of simulated annealing, we find that the procedure depends on $\boldsymbol{X} = \boldsymbol{a} + \boldsymbol{b}z$ at only one specific point, that is, the Metropolis algorithm, which is given by

$$
\texttt{status is} \begin{cases} \text{Acceptance} & \text{if } \Delta E(\boldsymbol{\mathcal{T}}_{i,l}, \boldsymbol{\mathcal{T}}_{i,l-1}, \boldsymbol{a} + \boldsymbol{b}r) + c_i \ln(\xi_{i,l}) < 0, \\[2mm] \text{Rejection} & \text{if } \Delta E(\boldsymbol{\mathcal{T}}_{i,l}, \boldsymbol{\mathcal{T}}_{i,l-1}, \boldsymbol{a} + \boldsymbol{b}r) + c_i \ln(\xi_{i,l}) \geq 0, \end{cases}
$$

where $\Delta E(\boldsymbol{\mathcal{T}}_{i,l}, \boldsymbol{\mathcal{T}}_{i,l-1}, \boldsymbol{a} + \boldsymbol{b}r) = E(\boldsymbol{\mathcal{T}}_{i,l}, \boldsymbol{a} + \boldsymbol{b}r) - E(\boldsymbol{\mathcal{T}}_{i,l-1}, \boldsymbol{a} + \boldsymbol{b}r)$, and $\boldsymbol{\mathcal{T}}_{i,l-1}, \boldsymbol{\mathcal{T}}_{i,l}$ respectively represent CP candidates before and after a transition to the neighborhood in the $l$-th iteration with a parameter $\xi_{i,l}$ for the $i$-th temperature $c_i$. Thus, to compute the region $\{r \in \mathbb{R} \,|\, \mathcal{S}_{\mathrm{SA}}(r) = \mathcal{S}_{\mathrm{SA}}(z)\}$ where the process of simulated annealing is identical, we need to condition on all results of the Metropolis algorithm at each temperature. This condition consists of multiple inequalities as follows:

$$
\{r \in \mathbb{R} \,|\, \mathcal{S}_{\mathrm{SA}}(r) = \mathcal{S}_{\mathrm{SA}}(z)\}
$$
$$
= \bigcap_{i=0}^{I_z} \bigcap_{l=1}^{L_z} \left\{ r \in \mathbb{R} \,\middle|\, \begin{cases} \Delta E(\boldsymbol{\mathcal{T}}_{i,l}, \boldsymbol{\mathcal{T}}_{i,l-1}, \boldsymbol{a} + \boldsymbol{b}r) + c_i \ln(\xi_{i,l}) < 0 & \text{if } \texttt{status} \text{ is Acceptance} \\ \Delta E(\boldsymbol{\mathcal{T}}_{i,l}, \boldsymbol{\mathcal{T}}_{i,l-1}, \boldsymbol{a} + \boldsymbol{b}r) + c_i \ln(\xi_{i,l}) \geq 0 & \text{if } \texttt{status} \text{ is Rejection} \end{cases} \right\}, \quad (22)
$$

where $I_z$ and $L_z$ denote the number of temperature updates and operations at each temperature for $z$, respectively.

We subsequently consider solving this inequality for $r$. The cost $\mathcal{C}\left(\boldsymbol{F}_{s+1:e}^{(d)}\right)$ used in the objective function $E$ can be expressed as a quadratic form of $\boldsymbol{X}$ such that

$$
\begin{aligned}
\mathcal{C}\left(\boldsymbol{F}_{s+1:e}^{(d)}\right) &= c_{\mathrm{sym}}^{(d)} \sum_{t=s+1}^{e} \left| F_t^{(d)} - \bar{F}_{s+1:e}^{(d)} \right|^2 \\
&= c_{\mathrm{sym}}^{(d)} \sum_{t=s+1}^{e} \left| \left(\boldsymbol{1}_{t:t} \otimes \boldsymbol{w}_M^{(d)}\right)^\top \boldsymbol{X} - \frac{1}{e-s}\left(\boldsymbol{1}_{s+1:e} \otimes \boldsymbol{w}_M^{(d)}\right)^\top \boldsymbol{X} \right|^2 \\
&= \boldsymbol{X}^\top C_{s+1:e}^{(d)} \boldsymbol{X},
\end{aligned}
$$

where

$$
C_{s+1:e}^{(d)} = c_{\mathrm{sym}}^{(d)} \sum_{t=s+1}^{e} \boldsymbol{u}_{s+1:e,t}^{(d)} \boldsymbol{u}_{s+1:e,t}^{(d)*\top} \in \mathbb{R}^{N \times N},
$$

$$
\boldsymbol{u}_{s+1:e,t}^{(d)} = \boldsymbol{1}_{t:t} \otimes \boldsymbol{w}_M^{(d)} - \frac{1}{e-s}\left(\boldsymbol{1}_{s+1:e} \otimes \boldsymbol{w}_M^{(d)}\right) \in \mathbb{C}^N,
$$

and $\boldsymbol{u}_{s+1:e,t}^{(d)*}$ is the complex conjugate of $\boldsymbol{u}_{s+1:e,t}^{(d)}$. Therefore, the objective function $E(\boldsymbol{\mathcal{T}}, \boldsymbol{a} + \boldsymbol{b}r)$ in (5) can be rewritten as follows:

$$
\begin{aligned}
E(\boldsymbol{\mathcal{T}}, \boldsymbol{a} + \boldsymbol{b}r) &= (\boldsymbol{a} + \boldsymbol{b}r)^\top \left( \sum_{d=0}^{D-1} \sum_{k=1}^{K^{(d)}+1} C_{\tau_{k-1}^{(d)}+1:\tau_k^{(d)}}^{(d)} \right) (\boldsymbol{a} + \boldsymbol{b}r) + \sum_{d=0}^{D-1} \beta^{(d)} K^{(d)} + \gamma K \\
&= e_2 r^2 + e_1 r + e_0, \quad (23)
\end{aligned}
$$

where

$$e_2 = \boldsymbol{b}^\top \sum_{d=0}^{D-1} \sum_{k=1}^{K^{(d)}+1} C_{\tau_{k-1}^{(d)}+1:\tau_k^{(d)}}^{(d)} \boldsymbol{b},$$

$$e_1 = 2\boldsymbol{a}^\top \sum_{d=0}^{D-1} \sum_{k=1}^{K^{(d)}+1} C_{\tau_{k-1}^{(d)}+1:\tau_k^{(d)}}^{(d)} \boldsymbol{b},$$

$$e_0 = \boldsymbol{a}^\top \sum_{d=0}^{D-1} \sum_{k=1}^{K^{(d)}+1} C_{\tau_{k-1}^{(d)}+1:\tau_k^{(d)}}^{(d)} \boldsymbol{a} + \sum_{d=0}^{D-1} \beta^{(d)} K^{(d)} + \gamma K.$$

Thus, the multiple inequalities in (22) can be easily solved after computing the coefficients in (23).

Note that the region $\{r \in \mathbb{R} \,|\, \mathcal{S}_{\text{pre-SA}}(r) = \mathcal{S}_{\text{pre-SA}}(z)\}$ can be computed similarly to (22) because the preliminary experiment is also based on the Metropolis algorithm. Therefore, this condition is formulated as

$$\{r \in \mathbb{R} \,|\, \mathcal{S}_{\text{pre-SA}}(r) = \mathcal{S}_{\text{pre-SA}}(z)\}$$

$$= \bigcap_{i=0}^{I_z^+} \bigcap_{l=1}^{L_z^+} \left\{ r \in \mathbb{R} \,\middle|\, \begin{cases} \Delta E(\boldsymbol{\mathcal{T}}_{i,l}^+, \boldsymbol{\mathcal{T}}_{i,l-1}^+, \boldsymbol{a}+\boldsymbol{b}r) + c_i^+ \ln(\xi_{i,l}^+) < 0 & \text{if } \texttt{status} \text{ is Acceptance} \\ \Delta E(\boldsymbol{\mathcal{T}}_{i,l}^+, \boldsymbol{\mathcal{T}}_{i,l-1}^+, \boldsymbol{a}+\boldsymbol{b}r) + c_i^+ \ln(\xi_{i,l}^+) \geq 0 & \text{if } \texttt{status} \text{ is Rejection} \end{cases} \right\},$$

where $\boldsymbol{\mathcal{T}}_{i,l}^+, c_i^+, \xi_{i,l}^+, I_z^+$, and $L_z^+$ in the preliminary experiment correspond to the respective parameters in (22), and $\boldsymbol{\mathcal{T}}_{i,0}^+ = \boldsymbol{\mathcal{T}}^{\text{init}}$.

Based on the above discussion, since the conditioning in (21) is represented as the intersection of multiple quadratic inequalities, the region $\mathcal{Z}^{\text{oc}}(\boldsymbol{a}+\boldsymbol{b}z)$ can be computed by solving them as

$$\mathcal{Z}^{\text{oc}}(\boldsymbol{a}+\boldsymbol{b}z) = \bigcup_{r=1}^{R_z} [L_{z(r)}^{\text{oc}}, U_{z(r)}^{\text{oc}}],$$

where, for $z$, $L_{z(r)}^{\text{oc}}$ and $U_{z(r)}^{\text{oc}}$ denote the lower and upper bounds of the $r$-th over-conditioned region, respectively, and $R_z$ represents the number of the intervals.

## C.2 Parametric Programming

Having derived $\mathcal{Z}^{\text{oc}}(\boldsymbol{a}+\boldsymbol{b}z)$ in the previous analysis, the region $\mathcal{Z}$ in Theorem 1 conditioned on the result of the algorithm $\mathcal{A}$ can be obtained using a computational method called parametric programming as follows:

$$\mathcal{Z} = \bigcup_{z \in \mathbb{R} \,|\, \mathcal{A}(\boldsymbol{a}+\boldsymbol{b}z) = \mathcal{A}(\boldsymbol{x})} \mathcal{Z}^{\text{oc}}(\boldsymbol{a}+\boldsymbol{b}z). \tag{24}$$

The line search method based on (24) for obtaining the region $\mathcal{Z}$ required for the computation of $p_k^{\text{selective}}$ is detailed in Algorithm 3. Furthermore, the overall procedure for computing selective $p$-values of the detected CP candidate locations is presented in Algorithm 4. An overview of the proposed search method is shown in Figure 9.

Watanabe & Suzuki (2021) proposed a selective inference method for model selection using simulated annealing in latent block models; however, this approach was limited to the specific algorithm and computed an approximated truncation region. In contrast, our proposed method can be applied to not only CP detection in the frequency domain which is the subject of this paper, but also a wide range of optimization problems solved using simulated annealing. Even in such a general case, we consider the over-conditioning based on the process of algorithm as in (21) and can obtain the "exact" truncation region using the parametric programming approach in (24).

---

**Algorithm 3** `compute_solution_path`

---

**Input:** Time seaquence $\boldsymbol{x}$, CP candidates $\boldsymbol{\mathcal{T}}$, CP candidate location $\tau_k$

1:  $z^{\mathrm{obs}} \leftarrow T_k(\boldsymbol{x})$ in (14)
2:  Compute $\boldsymbol{a}$ and $\boldsymbol{b}$ defined in Theorem 1
3:  Obtain $\mathcal{Z}^{\mathrm{oc}}(\boldsymbol{a} + \boldsymbol{b}z^{\mathrm{obs}})$ by (21)
4:  $S \leftarrow \mathcal{Z} \leftarrow \mathcal{Z}^{\mathrm{oc}}(\boldsymbol{a} + \boldsymbol{b}z^{\mathrm{obs}})$
5:  **while** $S^c \neq \emptyset$ **do**
6:      Obtain $\mathcal{Z}^{\mathrm{oc}}(\boldsymbol{a} + \boldsymbol{b}z)$ for $z \in S^c$ by (21)
7:      $S \leftarrow S \cup \mathcal{Z}^{\mathrm{oc}}(\boldsymbol{a} + \boldsymbol{b}z)$
8:      **if** $\mathcal{A}(\boldsymbol{a} + \boldsymbol{b}z) = \mathcal{A}(\boldsymbol{a} + \boldsymbol{b}z^{\mathrm{obs}})$ **then**
9:          $\mathcal{Z} \leftarrow \mathcal{Z} \cup \mathcal{Z}^{\mathrm{oc}}(\boldsymbol{a} + \boldsymbol{b}z)$
10:     **end if**
11: **end while**
**Output:** Truncation region $\mathcal{Z}$

---

---

**Algorithm 4** SI for Detected CP Candidate Locations

---

**Input:** Time seaquence $\boldsymbol{x}$

1:  $\boldsymbol{\mathcal{T}} \leftarrow \mathcal{A}(\boldsymbol{x})$
2:  Obtain $\boldsymbol{\tau}$ by (4)
3:  **for** $\tau_k \in \boldsymbol{\tau}$ **do**
4:      $\mathcal{Z} \leftarrow$ `compute_solution_path`$(\boldsymbol{x}, \boldsymbol{\mathcal{T}}, \tau_k)$
5:      Compute $p_k^{\mathrm{selective}}$ by (17)
6:  **end for**
**Output:** Detected CP candidate locations and the corresponding selective $p$-values $\left\{ \left( \tau_k, p_k^{\mathrm{selective}} \right) \right\}_{k=1}^{K}$

---

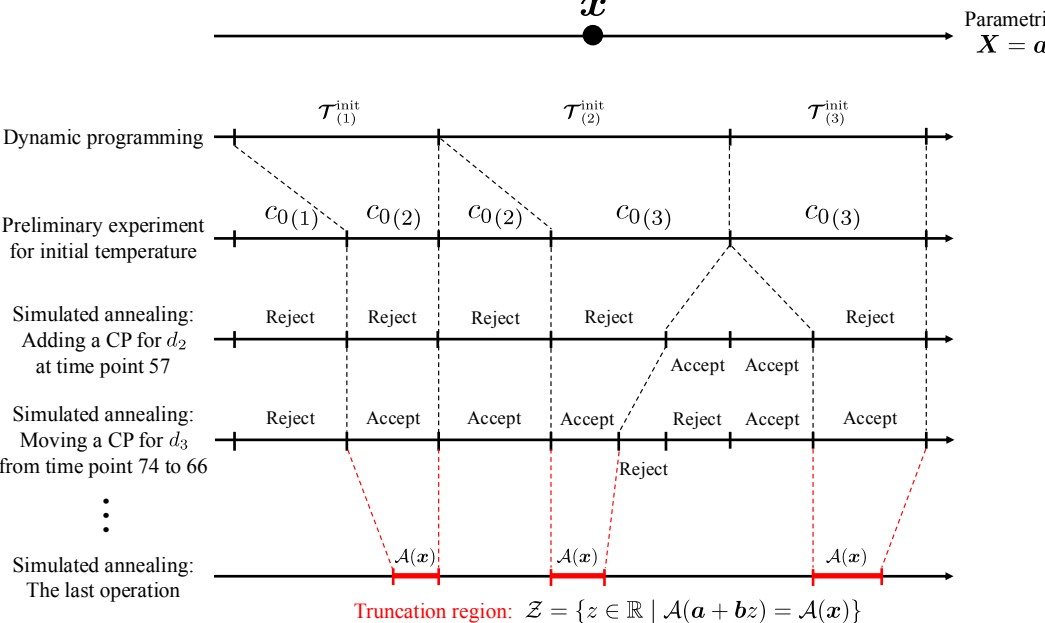

Figure 9: Schematic illustration of the proposed line search method for the identification of the truncation region. We first compute the over-conditioned region where the process of the algorithm $\mathcal{A}$ remains unchanged. Then, we identify the truncation region $\mathcal{Z}$ by removing the redundant conditioning using parametric programming.

# D   Details of Numerical Experiments

## D.1   Detailed Descriptions of Comparison Methods

In our experiments, we compared the proposed method (`Proposed`) with the following methods.

- `OC`: In this method, we consider $p$-values conditioned on the process of the algorithm $\mathcal{A}$. The $p$-value is computed by obtaining the over-conditioned region $\mathcal{Z}^{\mathrm{oc}}$ in (21) for the observed test statistic $T_k(\boldsymbol{x})$. This method is computationally efficient, however, its power is low due to over-conditioning.

- `OptSeg-SI-oc` (Duy et al., 2020): This method uses a $p$-value conditioned only on the event $\mathcal{S}_{\mathrm{DP}}$ in (21), that is, we consider conditioning on the process of dynamic programming algorithm.

- `OptSeg-SI` (Duy et al., 2020): This method removes over-conditionning from OptSeg-SI-oc, that is, the $p$-value is conditioned only on the result of dynamic programming.

- `Naive`: This method uses a conventional $p$-value without conditioning, which is computed as

$$p_k^{\mathrm{naive}} = \mathbb{P}_{\mathrm{H}_0,k}\left(T_k(\boldsymbol{X}) \geq T_k(\boldsymbol{x})\right).$$

- `Bonferroni`: This is a method to control the type I error rate by applying Bonferroni correction which is widely used as multiple testing correction. Since the number of all possible hypotheses is $m = (2^D - 1)(T - 1)$, the bonferroni $p$-value is computed by $p_k^{\mathrm{bonferroni}} = \min(1, m \cdot p_k^{\mathrm{naive}})$.

## D.2   Computational Time and Computer Resources

We measured the computational time of our proposed method for the synthetic data experiments presented in Section 5.2 and computed the medians for each settings. The results for the type I error rate and power experiments are shown in Figure 10. Panels (a) and (b) indicate that the computational time increases exponentially with the sequence length. In addition, the computational time becomes shorter as the signal intensity increases in panels (c) and (d). This may be because the results of hypothesis testing in the case of high intensity are more likely to be obvious, and the inference process can be terminated early. All numerical experiments were conducted on a computer with a 96-core 3.60GHz CPU and 512GB of memory.

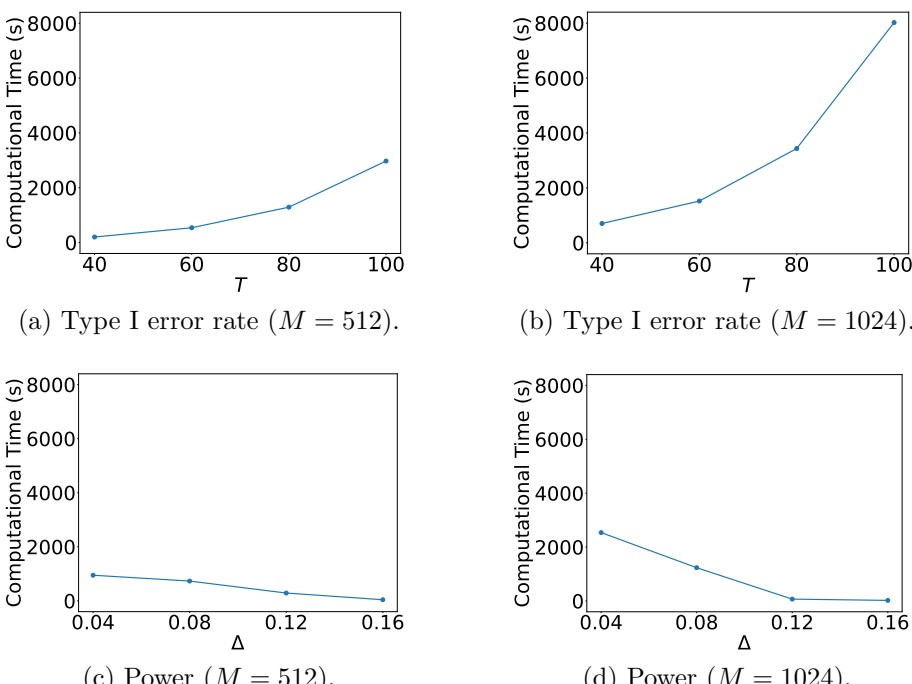

(a) Type I error rate ($M = 512$).      (b) Type I error rate ($M = 1024$).

(c) Power ($M = 512$).      (d) Power ($M = 1024$).

Figure 10: Computational time in the type I error rate and the power experiments.

### D.3 Robustness of Type I Error Rate Control

We evaluated the robustness of the `Proposed` in terms of the type I error rate control. For this purpose, we conducted experiments under three distinct noise conditions: (i) unknown noise variance, (ii) non-Gaussian noise, and (iii) correlated noise. The details of each experiment are given below.

**Unknown noise variance.** We generated 1000 null sequences in the same manner as the type I error rate experiment with known variance presented in Section 5.2. To estimate the variance $\sigma^2$ from the same data, we first applied CP candidate selection algorithm to identify the segments, and then computed the empirical variance of each segment for all frequencies. Since the estimated variance tended to be smaller than the true value, we adopted the maximum value for each frequency. Given $\hat{\sigma}_f$ as the average of the values, $\sigma$ could be estimated from the property of DFT as $\hat{\sigma} = \frac{\hat{\sigma}_f}{\sqrt{M}}$. The results for the significance levels $\alpha = 0.01, 0.05, 0.1$ are shown in Figure 11 and the `Proposed` still could properly control the type I error rate.

**Non-Gaussian noise.** We considered the case where the noise followed the five non-Gaussian distributions:

- `skewnorm`: Skew normal distribution family.

- `exponnorm`: Exponentially modified normal distribution family.

- `gennormsteep`: Generalized normal distribution family whose shape parameter $\beta$ is limited to be steeper than the normal distribution, i.e., $\beta < 2$.

- `gennormflat`: Generalized normal distribution family whose shape parameter $\beta$ is limited to be flatter than the normal distribution, i.e., $\beta > 2$.

- `t`: Student's t distribution family.

To generate sequences used in the experiment, we first obtained a noise distribution such that the 1-Wasserstein distance from the standard normal distribution $\mathcal{N}(0,1)$ was $\{0.01, 0.02, 0.03, 0.04\}$ in each aforementioned distribution family. Subsequently, we standardized the distribution to have a mean of 0 and a variance of 1. Then, we generated 1000 null sequences $\boldsymbol{x} = (x_1, \ldots, x_N)^\top$, where the mean vector was specified in the same manner as described in the type I error rate experiment with known variance, and the noise followed the obtained distribution, for $T = 60$. We applied hypothesis testing using the test statistic with $\sigma = 1$ for the detected CP candidate locations. The results for the significance levels $\alpha = 0.05$ are shown in Figure 12 and the `Proposed` could properly control the type I error rate for all non-Gaussian distributions.

**Correlated noise.** We generated 10000 null sequences $\boldsymbol{x} = (x_1, \ldots, x_N)^\top \sim \mathcal{N}(\boldsymbol{s}, \Sigma)$, where mean vector $\boldsymbol{s}$ was specified in the same manner as described in the type I error rate experiment with known variance, and covariance matrix $\Sigma$ was defined as $\Sigma = \sigma^2 \left( \rho^{|i-j|} \right)_{ij} \in \mathbb{R}^{N \times N}$ with $\sigma = 1$ and $\rho \in \{0.025, 0.05, 0.075, 0.1\}$, for $T = 60$. After CP candidate selection, we conducted the hypothesis testing using the test statistic with $\sigma = 1$. The results for the significance levels $\alpha = 0.01, 0.05, 0.1$ are shown in Figure 13. For weak covariance, the `Proposed` could properly control the type I error rate. However, the type I error rate could not be controlled with increasing noise correlation. This remains a challenge for future work.

### D.4 Sensitivity Study for the Penalty Parameter

This section presents a sensitivity study for the penalty parameter within the objective function in (5). As shown in Appendix A, the penalty parameter $\beta^{(d)}$ for each frequency $d$ can be theoretically derived using the BIC, leaving the penalty parameter $\gamma$ to be determined heuristically by users. Thus, we focus on the sensitivity study for $\gamma$ in this section. Specifically, we conducted the experiments for the type I error rate ($T = 60$) and power ($\Delta = 0.08$) on synthetic data as described in Section 5.2, while varying the hyperparameter $\kappa$ in (20) across the values $\{1.0, 1.5, 2.0, 2.5\}$. The results for the type I error rate and power experiments are shown in Figures 14 and 15, respectively. In Figure 14, the `Proposed` can control the type I error rate at the significance levels $\alpha = 0.01, 0.05, 0.1$ for all values of $\kappa$. Additionally, Figure 15 shows that the power of the `Proposed` is the highest of all methods, regardless of the settings of $\kappa$. In particular, for the experiment with $M = 512$, as the value of $\kappa$ increases, the power of the `Proposed` tends to be higher. This may be because a larger penalty makes distinct spectral changes more likely to be the subject of the hypothesis testing.

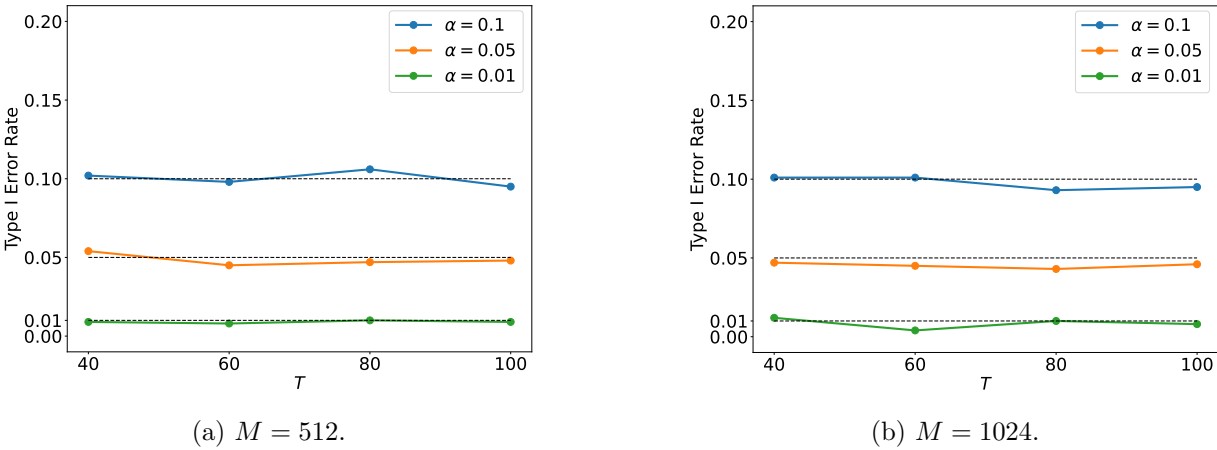

(a) $M = 512$.                                    (b) $M = 1024$.

Figure 11: Robustness of type I error control for estimated variance.

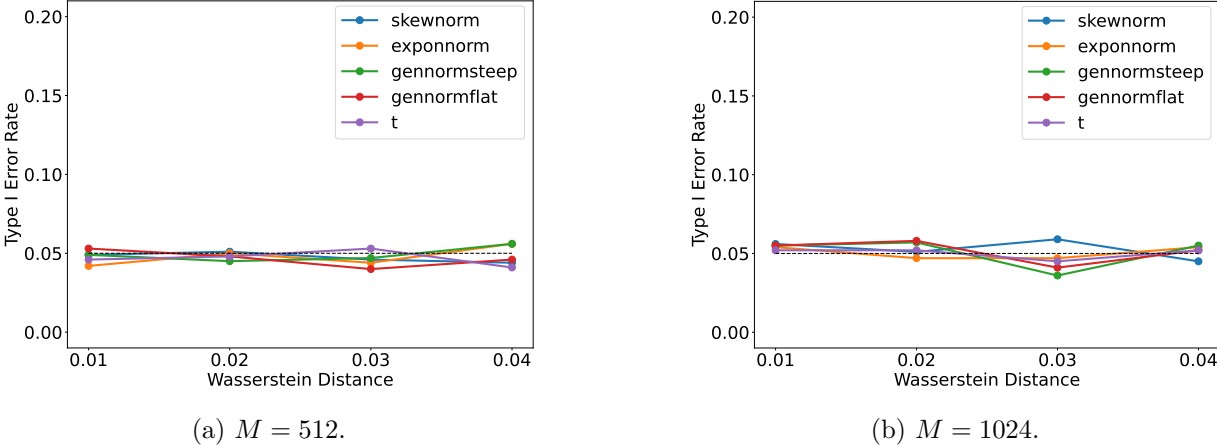

(a) $M = 512$.                                    (b) $M = 1024$.

Figure 12: Robustness of type I error control for non-Gaussian noise.

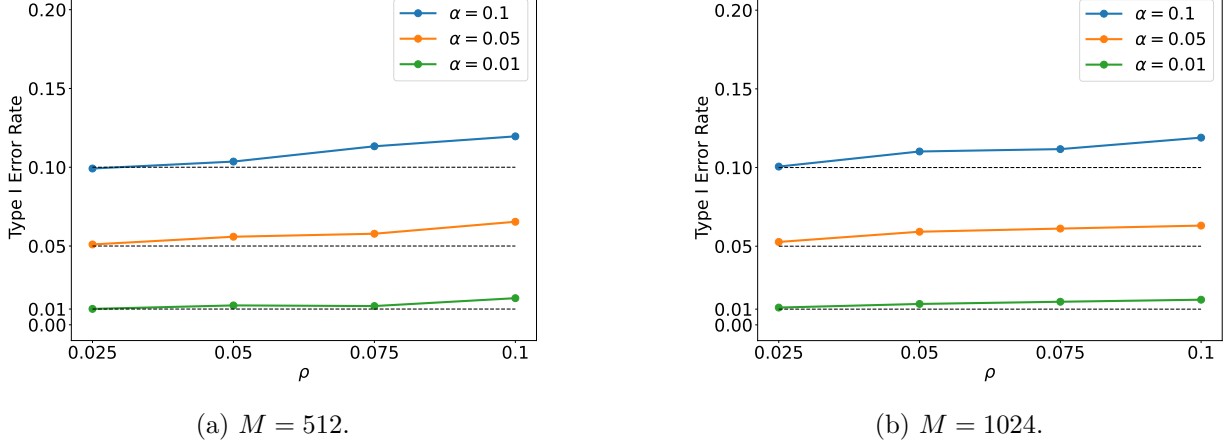

(a) $M = 512$.                                    (b) $M = 1024$.

Figure 13: Robustness of type I error control for correlation of noise.

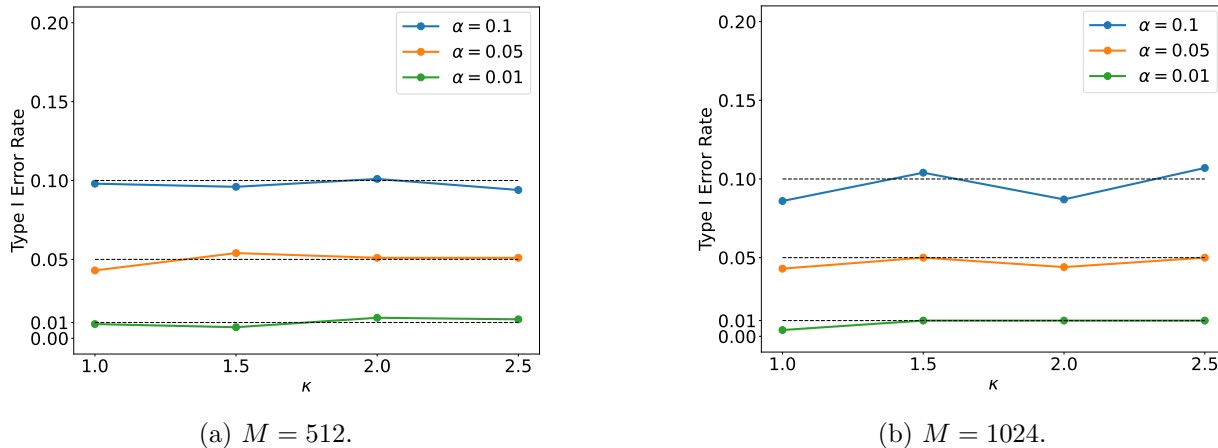

Figure 14: Sensitivity study of type I error rate control for the hyper-parameter $\kappa$ in the penalty parameter $\gamma$.

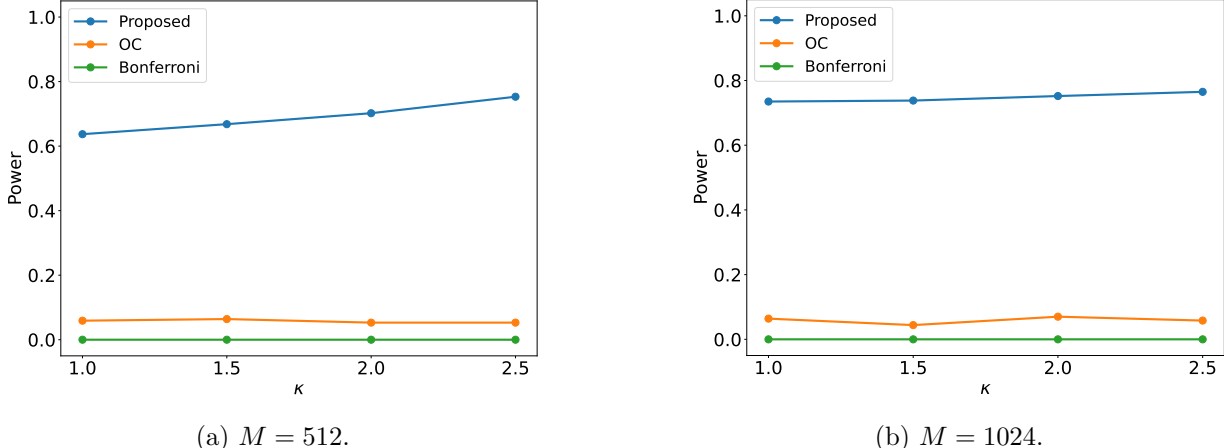

Figure 15: Sensitivity study of power for the hyper-parameter $\kappa$ in the penalty parameter $\gamma$.

## D.5 More Results on Real Data Experiments

Additional results for the signals of bearings 2, 3, and 4 before and after the spectral intensity enhancements occurred in the BPFO harmonics of bearing 1 are shown in Figures 16, 17, and 18. In panel (a) of each figure, the time variation of frequency spectra where CP candidate locations were falsely detected are shown for the period of 0.25–2.25 days when significant spectral changes in bearing 1 did not actually exist. The results indicate that the inferences using $p$-values of the `Proposed` and `OC` are valid. In panel (b), the time variations of the BPFO harmonics where CP candidate locations were correctly detected are presented for the period of 4–6 days (bearing 2), 4.75–6.75 days (bearing 3), and 4–6 days (bearing 4), respectively. In these cases, although the detection was delayed by several days relative to the occurrence of spectral amplification in bearing 1, the outer race fault signatures were successfully identified in all bearings using the `Proposed`.

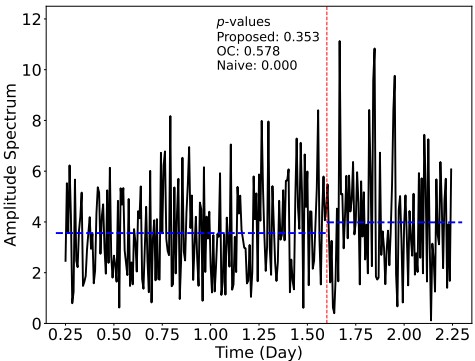

(a) Inference on a falsely detected CP candidate location for 3260 Hz (around the 14th harmonic) on 0.25–2.25 days.

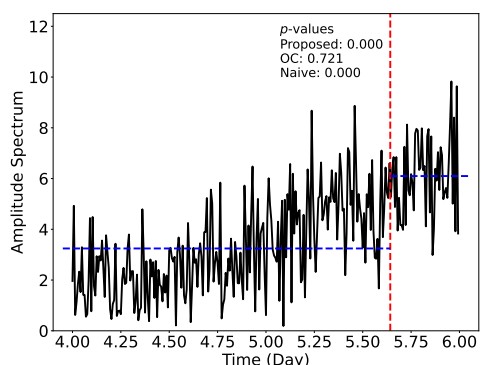

(b) Inference on a truly detected CP candidate location for 1420 Hz (the 6th harmonic) on 4–6 days.

Figure 16: Results of bearing 2. In panel (b), $p$-values were actually computed by considering CP candidates of the 6th harmonic, 3240 Hz, and 3460 Hz (around the 14th and 15th harmonics).

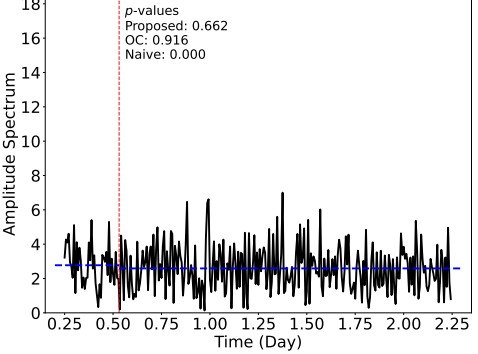

(a) Inference on a falsely detected CP candidate location for 3380 Hz (around the 14th harmonic) on 0.25–2.25 days.

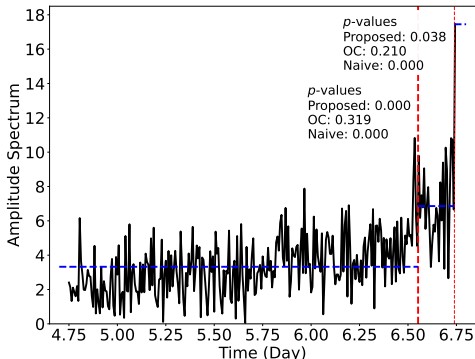

(b) Inferences on truly detected CP candidate locations for 1420 Hz (the 6th harmonic) on 4.75–6.75 days.

Figure 17: Results of bearing 3.

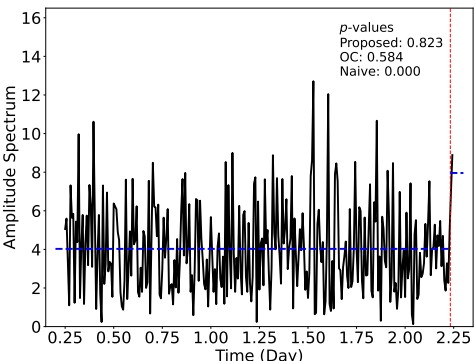

(a) Inference on a falsely detected CP candidate location for 3540 Hz (the 15th harmonic) on 0.25–2.25 days.

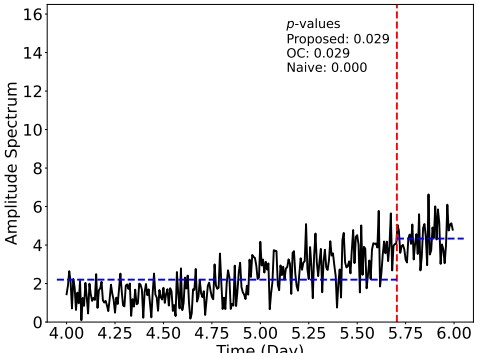

(b) Inference on a truly detected CP candidate location for 1420 Hz (the 6th harmonic) on 4–6 days.

Figure 18: Results of bearing 4. In panel (b), $p$-values were actually computed by considering not only a CP candidate of the 6th harmonic but also a CP candidate of 3440 Hz (around the 15th harmonic).

