# OpenReview forum: "Change Point Detection in the Frequency Domain with Statistical Reliability"
_TMLR — Accepted by TMLR_

### Review · Reviewer_Fed7 · 2025-02-28

**Summary Of Contributions:**

This paper introduces a statistical test for the significance of mean-shift change points in the frequency domain.

**Audience:**

Yes

**Claims And Evidence:**

Yes

**Requested Changes:**

Significant changes:
- The title and text should be changed to avoid confusion with time-series anomaly detection. If anomaly detection is an intended application, their relation to change points should be clearly discussed.
- Figure 2 should probably be removed.
- The authors should clearly point out and discuss the contributions of the paper, and improve the comparison to related work.

General Writing improvements:
- Some equations that should be numbered are not and some that are should not be, e.g. Why does eq. (4) have a dedicated equation? It seems to only be used for the definition of $K$.
- In general there is too much empty space caused by dedicated equations, which should probably be inline, e.g. (18) and (19).
- I am not familiar with the notation used in the definition of $c_{sym}^{(d)}$ and $A_n^{(d)}$. Usually, a case is followed by a statement that evaluates to true or false.
- The algorithms should be streamlined, e.g.
    - The nested conditions in Algorithm 1 can be combined without loosing expressiveness.
    - Algorithm 2 uses a mix of precise and imprecise notation, e.g. What is the "desired value"? Is "d" also uniformly sampled from $D^{init}$? Are the operations uniformly sampled? etc.
- The indices of $\tau$'s are hard to read. It seems, for every $\tau$ the authors only need its predecessor, its successor, and its associated frequency. If this is indeed the case, the equations could be simplified, e.g. by using $\tau_d^{pred}$, $\tau_d$, $\tau_d^{suc}$.
- The definitions of the two means in eq. (11) should be generalized for an arbitrary range $\bar{F}^{(d)}_{i:j}$ to remove redundancy.
- There are missing citations, e.g. "This additional conditioning on Q(X) is a standard approach for computational tractability in the SI literature."
- The cumulative distribution function shares a name with the Fourier Transform, which can be easily overlooked, since they both also come with multiple indices each.

Further points which would benefit from more explanation:
- What is the motivation for $\sigma^{-1}$ and $a_{k^{(d)}}$?
- It is not obvious, why eq. (12) holds.
- The step "[...] $Z = T_k(X)$ [...] that satisfy $X = a + bZ$ [...]" is not clear.
- Theorem 2.

**Strengths And Weaknesses:**

- The title and abstract of the paper are misleading, as the paper is not about anomaly detection in time series, but rather about the detection of a specific type of change points.
- Without any substantial experience in the field, it is not clear what the contribution of the paper is. The authors point to section 4 multiple times for the novelty, but most results are "standard approaches" or results from other papers according to the paper.
- It is not obvious how the works discussed in the related work section relate to the proposed methodology.
- Figure 2 is essentially contained in Figure 1.
- The writing is inconsistent and the notation is overly complicated, making the paper very difficult to follow.

---

> ### Author Response · Authors · 2025-03-31
> **Our Replies to Reviewer Fed3 (part 1)**
>
> We sincerely appreciate your time in reviewing our paper and providing constructive feedback. We especially thank you for careful reading and suggestions to improve the clarity of the writing.
>
> > The title and text should be changed to avoid confusion with time-series anomaly detection. If anomaly detection is an intended application, their relation to change points should be clearly discussed.
>
> We agree with your opinion. We have replaced the word "anomaly" with "change point (CP)" or "change" throughout the paper, including the title and abstract (we are currently checking with the Action Editor to confirm whether changing the title is allowed).
>
> > Figure 2 should probably be removed.
>
> Following the suggestion, we removed Figure 2.
>
> > The authors should clearly point out and discuss the contributions of the paper, and improve the comparison to related work.
>
> The main contribution of this study is the development of a statistical testing method, based on the framework of Selective Inference (SI), to quantify the statistical significance of change point (CP) detection in the frequency domain. While CP detection in the frequency domain is an important problem in engineering and several CP detection algorithms have been proposed, to the best of our knowledge, there has been no existing method that properly quantifies the statistical significance of detected CPs in the form of p-values. By making effective use of the recently advancing SI framework, we provide a solution to this unresolved issue—this constitutes our main contribution.
>
> Regarding comparisons with related works, since no existing method can correctly compute p-values, we consider the following baselines in our numerical experiments: Naive, Bonferroni, OC, OptSeg-SI, and OptSeg-SI-oc. The Naive approach does not account for the fact that the CPs are detected based on the data, and thus fails to properly evaluate statistical significance due to selection bias (e.g., it cannot control the type I error under a fixed significance level). Bonferroni, on the other hand, is a multiple testing correction method that adjusts p-values by assuming tests over all possible combinations of CPs. Since the number of combinations increases exponentially, Bonferroni becomes overly conservative, resulting in almost zero detection power. The methods OC, OptSeg-SI, and OptSeg-SI-oc are not existing methods but are included as ablation studies.
>
> In summary, our contribution is not the proposal of a new CP detection method, but the development of a statistical testing method that can be applied to existing CP detection algorithms. We acknowledge the reviewer's concern that understanding the context of statistical testing may be challenging for general machine learning researchers. In the revised version, we have added further explanations to clarify the contributions of this study.

---

> ### Author Response · Authors · 2025-03-31
> **Our Replies to Reviewer Fed3 (part 2)**
>
> >  Some equations that should be numbered are not and some that are should not be, e.g. Why does eq. (4) have a dedicated equation? It seems to only be used for the definition of K.
>
> Equation (4) appears in line 2 of Algorithm 3. We have checked that all numbered equations are referenced at least once in the paper.
>
> > In general there is too much empty space caused by dedicated equations, which should probably be inline, e.g. (18) and (19).
>
> Thank you for the suggestion. In the revised version, we have significantly updated Section 4.2, so the issue has now been addressed.
>
> > I am not familiar with the notation used in the definition of $c_{sym}^{(d)}$ and $A_n^{(d)}$. Usually, a case is followed by a statement that evaluates to true or false.
>
> Following the comment, we have revised these expressions using case notation with ``if'' statements.
>
> > The algorithms should be streamlined, e.g., ...
>
> We have fixed several parts of Algorithms 1, 2, 3, and 4.
>
> > The indices of $\tau$s are hard to read. It seems, for every $\tau$, the authors only need its predecessor, its successor, and its associated frequency. If this is indeed the case, the equations could be simplified, e.g. by using $\tau_d^{pred}$, $\tau_{d}$, $\tau_{d}^{suc}$.
>
> In the initial draft, we distinguished multiple change points in the notation, but following the suggestion, we revised the description to focus on a single change point. We hope this change helps make it easier for readers to understand.
>
> > The definitions of the two means in eq. (11) should be generalized for an arbitrary range ${\mathcal F}_{i:j}^{(d)}$ to remove redundancy.
>
> Following the suggestion, we generalized and redefined them.
>
> > There are missing citations, e.g. "This additional conditioning on Q(X) is a standard approach for computational tractability in the SI literature."
>
> We have added the reference "loftus2015selective" for this part.
>
> > The cumulative distribution function shares a name with the Fourier Transform, which can be easily overlooked, since they both also come with multiple indices each.
>
> Thank you for pointing this out. In the revised version, we have significantly updated Section 4.2, so the issue has now been addressed.
>
> > What is the motivation for $\sigma^{-1}$ and $a_{k^{(d)}}$?
>
> $\sigma^{-1}$ is just for standardization, and $a_{len}^{(d)}$  is a scaling factor for correcting the segment lengths. In the revised manuscript, we have added this explanation.
>
> > It is not obvious, why eq. (12) holds.
>
> We have added three lines of additional equations to help readers understand equation (12).
>
> > The step "[...] $Z=T_k(X)$ [...] that satisfy $X = a + b Z$  [...]" is not clear.
>
> In the revised version, we have significantly updated Section 4.2, so the issue has now been addressed.
>
> > Theorem 2.
>
> In the revised version, we have significantly updated Section 4.2, so the issue has now been addressed.

---

### Review · Reviewer_nMys · 2025-03-08

**Summary Of Contributions:**

This paper presents a multi-resolution change point (CP) detection approach, which incorporates three main ideas:

1. Performing CP detection simultaneously at multiple frequencies.
1. Utilizing the dynamic programming (DP) approach for individual CP detection.
1. Consolidating the individual CPs using a simulated annealing approach.

Additionally, the authors introduce a method for computing the confidence of the identified CPs using selective inference (SI).

**Audience:**

Yes

**Broader Impact Concerns:**

None.

**Claims And Evidence:**

No

**Requested Changes:**

See the weakness.

**Strengths And Weaknesses:**

While the techniques used are known, the proposed multi-resolution CP detection method described in the first half of the paper appears solid and useful.

Unlike the first half, which presents the CP detection approach itself, Section 4, which focuses on the SI approach, is somewhat vague and difficult to follow.

The procedure seems to be based on the assumption that the observations are i.i.d. with a constant variance $\sigma^2$ in Eq. (1), the validity of which should be verified. The underlying intuition appears to be that the quadratic statistic in Eq. (11) should follow a Chi-squared-type distribution. However, no clear procedure is provided especially for determining the linear transformation defined in Eq. (17) in a data-driven fashion.

Although the authors present a real-world data analysis example in Section 5.3, the details of the analysis are not clearly described, which may leave many readers confused.

Overall, while the first half of the paper presents a useful framework for multi-resolution CP detection, the second half on SI does not seem to add significant value.

---

> ### Author Response · Authors · 2025-03-31
> **Our Replies to Reviewer nMys**
>
> We sincerely appreciate your time in reviewing our paper and providing constructive feedback.
>
> > Section 4, which focuses on the SI approach, is somewhat vague and difficult to follow.
>
> A similar comment was made by another reviewer (Fed7). We completely rewrote Section 4 using simpler expressions.
>
> > The procedure seems to be based on the assumption that the observations are i.i.d. with a constant variance $\sigma^2$ in Eq. (1), the validity of which should be verified. The underlying intuition appears to be that the quadratic statistic Eq. (11) should follow a Chi-squared-type distribution.
>
> Yes. This is a limitation of current SI. Unless the covariance matrix is a scalar multiple of the identity matrix, the sampling distribution of the test statistic does not follow truncated chi-squared distribution. To investigate the performance when this assumption is not satisfied, we conducted numerical experiments in Appendix D.3 for the case where the covariance matrix has non-zero off-diagonal elements. The results indicate that the type I error rate is controlled when the off-diagonal elements are sufficiently small compared to the diagonal ones, but not when the off-diagonal elements are relatively large. In the revised version, we have added this point under the Limitations section.
>
> > However, no clear procedure is provided especially for determining the linear transformation defined in Eq. (17) in a data-driven fashion.
>
> If $a$, $b$, and $Z$ are defined as in equations (18) and (19), the equivalence between equation (17) and equation (16) is shown in the proof of Theorem 1. Equation (17) does not imply a linear transformation; rather, it constructively shows that the set of data satisfying the selection event in equation (16) lies in a one-dimensional subspace within the N-dimensional space. In the revised version, we have added an explanation to clarify this point.
>
> > Although the authors present a real-world data analysis example in Section 5.3, the details of the analysis are not clearly described, which may leave many readers confused.
>
> In Section 5.3, we have added an explanation regarding the real data experiments.

---

### Review · Reviewer_JRxG · 2025-03-18

**Summary Of Contributions:**

This paper tackles the problem of change point (CP) detection in the frequency domain to effectively identify anomalies in complex systems.
The core contribution of this work is the application of the Selective Inference (SI) framework to the detection of CP in the frequency domain.
The proposed SI method quantifies the statistical significance of detected CPs in the frequency domain using $p$-values, ensuring that the detected changes reflect genuine structural shifts in the target system rather than random noise.

The paper addresses two major technical challenges to achieve this. First, it extends the existing SI framework to the frequency domain by appropriately utilizing the properties of the discrete Fourier transform (DFT).
Second, it develops an SI method that provides valid p-values for CPs where changes occur across multiple frequencies.
The proposed method consists of two stages: a heuristic selection of CP candidates in the frequency domain, followed by a quantification of the statistical significance of each candidate using the SI framework.

**Audience:**

Yes

**Broader Impact Concerns:**

None.

**Claims And Evidence:**

Yes

**Requested Changes:**

I don't really have changes to request, the points below should rather be seen as potential improvements that the authors could (or not) integrate in a future revision of the paper:

* The authors could discuss the limitations of using a heuristic algorithm (simulated annealing) for CP candidate selection in more detail and potentially explore or suggest alternative or improved heuristic approaches. They could also explain in a few words why the dynamic programming approach that is used for each frequency cannot be extended to the cross-frequency problem.
* It would be helpful to provide more insight into the impact of hyperparameters $\beta(d)$ and $\gamma$ on the results. In this regard, a sensitivity study included in the paper's main body could have been a plus.

**Strengths And Weaknesses:**

**Strengths:**

* The paper presents a novel approach by extending statistically significant CP detection using Selective Inference (SI) to the frequency domain, an area that is less explored than the time domain.
* The proposed method quantifies the statistical significance of detected CPs using $p$-values, which helps in distinguishing real changes from noise.
* The experimental results on synthetic and real-world data demonstrate the reliability of the proposed method in identifying true CPs while controlling the type I error rate below the specified significance level.
* The paper examines the robustness of the proposed method to unknown noise variance and non-Gaussian noise, showing promising results under these conditions.

**Weakness:**

* The selection of CP candidates in the first stage relies on a heuristic algorithm (simulated annealing) due to the complexity of the combinatorial optimization problem. While simulated annealing is a practical method for finding approximate solutions, it does not guarantee obtaining the globally optimal set of CPs.

---

> ### Author Response · Authors · 2025-03-31
> **Our Replies to Reviewer JRxG**
>
> We sincerely appreciate your time in reviewing our paper and providing constructive feedback.
>
> > The authors could discuss the limitations of using a heuristic algorithm (simulated annealing) for CP candidate selection in more detail and potentially explore or suggest alternative or improved heuristic approaches.
>
> The problem of identifying partially shared change points (CPs) across time series with different frequencies is a combinatorial optimization problem, making it difficult to obtain a globally optimal solution and thus necessitating the use of meta-heuristic methods to find an approximate solution (as will be discussed later, while the problem of finding the optimal CPs in a single time series can be solved using dynamic programming (DP), identifying partially shared CPs across multiple time series cannot be addressed by DP). Simulated Annealing (SA) is one well-known and effective meta-heuristic, and we adopted it in this paper as an example. Although it would be possible to use other meta-heuristics, we believe that they would not significantly improve detection accuracy compared to SA. We emphasize that the main contribution of our study does not lie in the choice of a specific meta-heuristic, but rather in constructing a theoretically valid statistical test using Selective Inference (SI) that properly accounts for the use of a meta-heuristic. The proposed SI framework is applicable even when other heuristics are employed, as long as their selection events can be characterized, as we have done in this paper for SA. In the revised version, we have added an explanation to clarify this point.
>
> > They could also explain in a few words why the dynamic programming approach that is used for each frequency cannot be extended to the cross-frequency problem.
>
> As is well-known, optimal CPs for a single time series can be efficiently obtained using DP. However, when it comes to the problem of identifying shared CPs across different frequencies, DP is not anymore applicable. This stems from the fact that we cannot construct a Bellman equation for this optimization problem, i.e., additional information that is not retained in the solutions of subproblems may be required for later-stage decisions. We have added an explanation addressing this comment in Section 3.2.
>
> > It would be helpful to provide more insight into the impact of hyper-parameters $\beta$ and $\gamma$ on the result. In this regard, a sensitivity study included in the paper's main body could have been a plus.
>
> In the revised version, we have added Appendix D.4, which presents a sensitivity analysis of the hyper-parameters. Among the two hyper-parameters, $\beta^{(d)}$ and $\gamma$, the former can be uniquely determined based on the Bayesian Information Criterion (BIC), so the sensitivity analysis was conducted with respect to $\gamma$. We emphasize that, regardless of the choice of hyper-parameters, the validity of the statistical test based on the SI framework remains satisfied.
>
> > The selection of CP candidates in the first stage relies on a heuristic algorithm (simulated annealing) due to the complexity of the combinatorial optimization problem. While simulated annealing is a practical method for finding approximate solutions, it does not guarantee obtaining the globally optimal set of CPs.
>
> As mentioned above, since it is difficult to find a globally optimal solution to the problem of identifying shared CPs across different frequencies, we must rely on meta-heuristics such as SA. This reliance is inevitable due to the complexity of the problem, and overcoming this challenge is not the objective of this study. The main contribution of this work lies in conducting valid statistical inference that properly accounts for the use of a specific meta-heuristic to obtain an approximate solution. In the revised version, we have added an explanation to clarify this point.

---

### Author Response · Authors · 2025-03-31
**Revised Manuscript**

We revised the paper based on the feedback from the three reviewers. The revised parts are marked in red (if an entire section was revised, the section title is marked in red). The main changes are as follows:

[1] Following the suggestion from a reviewer, we have replaced the word "anomaly" with "change point (CP)" or "change" throughout the paper, including the title and abstract.

[2] To clarify the contributions of this study, we have added a paragraph on our contributions just before the Related Works section in Section 1.

[3] In Section 3.2, we added an explanation that, unlike the CP detection problem for uni-variate time series, dynamic programming (DP) cannot be used in the optimization problem for identifying CPs common across multiple frequency domains, along with the reason for this limitation.

[4] Following the suggestion from a Reviewer, we removed Figure 2 of the first draft.

[5] Following the suggestion from a reviewer, we revised the presentation of Algorithms 1, 2, 3, and 4.

[6] In the explanation of statistical testing for CPs in Section 4, we simplified the description by adopting a notation that does not distinguish between multiple CPs.

[7] Since multiple reviewers pointed out that Section 4 in the first draft was difficult to follow, we significantly revised Section 4 in the revised version.

[8] To investigate the sensitivity to hyper-parameters, we conducted additional numerical experiments and reported the results in Appendix D.4.

[9] In Section 5.3, we have added a detailed description of the real data experiments.

[10] In Section 6, we have added a discussion of the current limitations of Selective Inference.

---

> ### Comment · Action_Editor_SDzj · 2025-04-01
>
> Dear Reviewers,
>
> The authors have provided a revision and response to your comments. Please take a look and engage in a discussion to clarify any further concerns before you make your recommendation.
>
> Thanks,
> AE

---

### Decision · Action_Editor_SDzj · 2025-04-26

**Recommendation:** Accept as is

**Comment:**

This work presents a novel statistical testing method for change-point detection in the frequency domain, which is supported by experiments on toy and real data. The authors have revised the paper, especially for readability, according to the reviewers' feedback and majority are positive about the paper. The revised paper clearly meets the criteria for acceptance. The authors can consider further improving readability for a more general machine learning audience.

**Audience:**

While on the more statistical side, change-point detection is an important practical problem and being able to quantify the statistical significance is a useful contribution. It should be of interest to at least some in the community.

**Claims And Evidence:**

The paper claims to propose a statistical testing method for change-point detection in the frequency domain. The revised paper clarifies the contributions of the work, being the statistical testing method, and also updates the title to reflect the contribution more clearly. The theory and experiments on synthetic and real data support the claims.